# Score-Based Generative Modeling through Stochastic Differential Equations

**Yang Song***
Stanford University
yangsong@cs.stanford.edu

**Jascha Sohl-Dickstein**
Google Brain
jaschasd@google.com

**Diederik P. Kingma**
Google Brain
durk@google.com

**Abhishek Kumar**
Google Brain
abhishk@google.com

**Stefano Ermon**
Stanford University
ermon@cs.stanford.edu

**Ben Poole**
Google Brain
pooleb@google.com

## Abstract

Creating noise from data is easy; creating data from noise is generative modeling. We present a stochastic differential equation (SDE) that smoothly transforms a complex data distribution to a known prior distribution by slowly injecting noise, and a corresponding reverse-time SDE that transforms the prior distribution back into the data distribution by slowly removing the noise. Crucially, the reverse-time SDE depends only on the time-dependent gradient field (a.k.a., score) of the perturbed data distribution. By leveraging advances in score-based generative modeling, we can accurately estimate these scores with neural networks, and use numerical SDE solvers to generate samples. We show that this framework encapsulates previous approaches in score-based generative modeling and diffusion probabilistic modeling, allowing for new sampling procedures and new modeling capabilities. In particular, we introduce a predictor-corrector framework to correct errors in the evolution of the discretized reverse-time SDE. We also derive an equivalent neural ODE that samples from the same distribution as the SDE, but additionally enables exact likelihood computation, and improved sampling efficiency. In addition, we provide a new way to solve inverse problems with score-based models, as demonstrated with experiments on class-conditional generation, image inpainting, and colorization. Combined with multiple architectural improvements, we achieve record-breaking performance for unconditional image generation on CIFAR-10 with an Inception score of 9.89 and FID of 2.20, a competitive likelihood of 2.99 bits/dim, and demonstrate high fidelity generation of $1024 \times 1024$ images for the first time from a score-based generative model.

## 1 Introduction

Two successful classes of probabilistic generative models involve sequentially corrupting training data with slowly increasing noise, and then learning to reverse this corruption in order to form a generative model of the data. *Score matching with Langevin dynamics* (SMLD) (Song & Ermon, 2019) estimates the *score* (*i.e.*, the gradient of the log probability density with respect to data) at each noise scale, and then uses Langevin dynamics to sample from a sequence of decreasing noise scales during generation. *Denoising diffusion probabilistic modeling* (DDPM) (Sohl-Dickstein et al., 2015; Ho et al., 2020) trains a sequence of probabilistic models to reverse each step of the noise corruption, using knowledge of the functional form of the reverse distributions to make training tractable. For continuous state spaces, the DDPM training objective implicitly computes scores at each noise scale. We therefore refer to these two model classes together as *score-based generative models*.

Score-based generative models, and related techniques (Bordes et al., 2017; Goyal et al., 2017; Du & Mordatch, 2019), have proven effective at generation of images (Song & Ermon, 2019; 2020; Ho et al., 2020), audio (Chen et al., 2020; Kong et al., 2020), graphs (Niu et al., 2020), and shapes (Cai

---

*Work partially done during an internship at Google Brain.

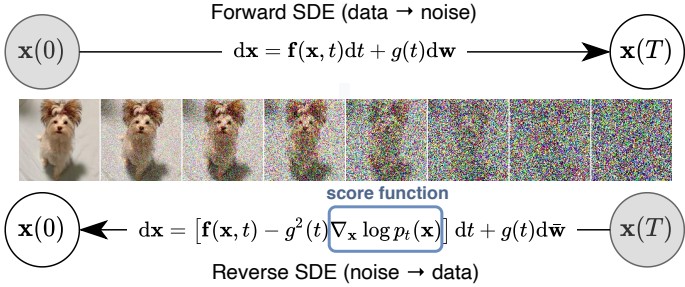

Figure 1: **Solving a reverse-time SDE yields a score-based generative model.** Transforming data to a simple noise distribution can be accomplished with a continuous-time SDE. This SDE can be reversed if we know the score of the distribution at each intermediate time step, $\nabla_{\mathbf{x}} \log p_t(\mathbf{x})$.

et al., 2020). To enable new sampling methods and further extend the capabilities of score-based generative models, we propose a unified framework that generalizes previous approaches through the lens of stochastic differential equations (SDEs).

Specifically, instead of perturbing data with a finite number of noise distributions, we consider a continuum of distributions that evolve over time according to a diffusion process. This process progressively diffuses a data point into random noise, and is given by a prescribed SDE that does not depend on the data and has no trainable parameters. By reversing this process, we can smoothly mold random noise into data for sample generation. Crucially, this reverse process satisfies a reverse-time SDE (Anderson, 1982), which can be derived from the forward SDE given the score of the marginal probability densities as a function of time. We can therefore approximate the reverse-time SDE by training a time-dependent neural network to estimate the scores, and then produce samples using numerical SDE solvers. Our key idea is summarized in Fig. 1.

Our proposed framework has several theoretical and practical contributions:

**Flexible sampling and likelihood computation:** We can employ any general-purpose SDE solver to integrate the reverse-time SDE for sampling. In addition, we propose two special methods not viable for general SDEs: (i) Predictor-Corrector (PC) samplers that combine numerical SDE solvers with score-based MCMC approaches, such as Langevin MCMC (Parisi, 1981) and HMC (Neal et al., 2011); and (ii) deterministic samplers based on the probability flow ordinary differential equation (ODE). The former *unifies and improves* over existing sampling methods for score-based models. The latter allows for *fast adaptive sampling* via black-box ODE solvers, *flexible data manipulation* via latent codes, a *uniquely identifiable encoding*, and notably, *exact likelihood computation*.

**Controllable generation:** We can modulate the generation process by conditioning on information not available during training, because the conditional reverse-time SDE can be efficiently estimated from *unconditional* scores. This enables applications such as class-conditional generation, image inpainting, colorization and other inverse problems, all achievable using a single unconditional score-based model without re-training.

**Unified framework:** Our framework provides a unified way to explore and tune various SDEs for improving score-based generative models. The methods of SMLD and DDPM can be amalgamated into our framework as discretizations of two separate SDEs. Although DDPM (Ho et al., 2020) was recently reported to achieve higher sample quality than SMLD (Song & Ermon, 2019; 2020), we show that with better architectures and new sampling algorithms allowed by our framework, the latter can catch up—it achieves new state-of-the-art Inception score (9.89) and FID score (2.20) on CIFAR-10, as well as high-fidelity generation of $1024 \times 1024$ images for the first time from a score-based model. In addition, we propose a new SDE under our framework that achieves a likelihood value of 2.99 bits/dim on uniformly dequantized CIFAR-10 images, setting a new record on this task.

## 2 BACKGROUND

### 2.1 DENOISING SCORE MATCHING WITH LANGEVIN DYNAMICS (SMLD)

Let $p_\sigma(\tilde{\mathbf{x}} \mid \mathbf{x}) := \mathcal{N}(\tilde{\mathbf{x}}; \mathbf{x}, \sigma^2 \mathbf{I})$ be a perturbation kernel, and $p_\sigma(\tilde{\mathbf{x}}) := \int p_{\text{data}}(\mathbf{x}) p_\sigma(\tilde{\mathbf{x}} \mid \mathbf{x}) d\mathbf{x}$, where $p_{\text{data}}(\mathbf{x})$ denotes the data distribution. Consider a sequence of positive noise scales $\sigma_{\min} = \sigma_1 < \sigma_2 < \cdots < \sigma_N = \sigma_{\max}$. Typically, $\sigma_{\min}$ is small enough such that $p_{\sigma_{\min}}(\mathbf{x}) \approx p_{\text{data}}(\mathbf{x})$, and $\sigma_{\max}$ is

large enough such that $p_{\sigma_{\max}}(\mathbf{x}) \approx \mathcal{N}(\mathbf{x}; \mathbf{0}, \sigma_{\max}^2 \mathbf{I})$. Song & Ermon (2019) propose to train a Noise Conditional Score Network (NCSN), denoted by $\mathbf{s}_{\boldsymbol{\theta}}(\mathbf{x}, \sigma)$, with a weighted sum of denoising score matching (Vincent, 2011) objectives:

$$\boldsymbol{\theta}^* = \arg\min_{\boldsymbol{\theta}} \sum_{i=1}^{N} \sigma_i^2 \mathbb{E}_{p_{\text{data}}(\mathbf{x})} \mathbb{E}_{p_{\sigma_i}(\tilde{\mathbf{x}}|\mathbf{x})} \big[ \|\mathbf{s}_{\boldsymbol{\theta}}(\tilde{\mathbf{x}}, \sigma_i) - \nabla_{\tilde{\mathbf{x}}} \log p_{\sigma_i}(\tilde{\mathbf{x}} \mid \mathbf{x}) \|_2^2 \big]. \tag{1}$$

Given sufficient data and model capacity, the optimal score-based model $\mathbf{s}_{\boldsymbol{\theta}^*}(\mathbf{x}, \sigma)$ matches $\nabla_{\mathbf{x}} \log p_{\sigma}(\mathbf{x})$ almost everywhere for $\sigma \in \{\sigma_i\}_{i=1}^N$. For sampling, Song & Ermon (2019) run $M$ steps of Langevin MCMC to get a sample for each $p_{\sigma_i}(\mathbf{x})$ sequentially:

$$\mathbf{x}_i^m = \mathbf{x}_i^{m-1} + \epsilon_i \mathbf{s}_{\boldsymbol{\theta}^*}(\mathbf{x}_i^{m-1}, \sigma_i) + \sqrt{2\epsilon_i} \mathbf{z}_i^m, \quad m = 1, 2, \cdots, M, \tag{2}$$

where $\epsilon_i > 0$ is the step size, and $\mathbf{z}_i^m$ is standard normal. The above is repeated for $i = N, N - 1, \cdots, 1$ in turn with $\mathbf{x}_N^0 \sim \mathcal{N}(\mathbf{x} \mid \mathbf{0}, \sigma_{\max}^2 \mathbf{I})$ and $\mathbf{x}_i^0 = \mathbf{x}_{i+1}^M$ when $i < N$. As $M \to \infty$ and $\epsilon_i \to 0$ for all $i$, $\mathbf{x}_1^M$ becomes an exact sample from $p_{\sigma_{\min}}(\mathbf{x}) \approx p_{\text{data}}(\mathbf{x})$ under some regularity conditions.

## 2.2 DENOISING DIFFUSION PROBABILISTIC MODELS (DDPM)

Sohl-Dickstein et al. (2015); Ho et al. (2020) consider a sequence of positive noise scales $0 < \beta_1, \beta_2, \cdots, \beta_N < 1$. For each training data point $\mathbf{x}_0 \sim p_{\text{data}}(\mathbf{x})$, a discrete Markov chain $\{\mathbf{x}_0, \mathbf{x}_1, \cdots, \mathbf{x}_N\}$ is constructed such that $p(\mathbf{x}_i \mid \mathbf{x}_{i-1}) = \mathcal{N}(\mathbf{x}_i; \sqrt{1 - \beta_i} \mathbf{x}_{i-1}, \beta_i \mathbf{I})$, and therefore $p_{\alpha_i}(\mathbf{x}_i \mid \mathbf{x}_0) = \mathcal{N}(\mathbf{x}_i; \sqrt{\alpha_i} \mathbf{x}_0, (1 - \alpha_i) \mathbf{I})$, where $\alpha_i := \prod_{j=1}^{i}(1 - \beta_j)$. Similar to SMLD, we can denote the perturbed data distribution as $p_{\alpha_i}(\tilde{\mathbf{x}}) := \int p_{\text{data}}(\mathbf{x}) p_{\alpha_i}(\tilde{\mathbf{x}} \mid \mathbf{x}) \mathrm{d}\mathbf{x}$. The noise scales are prescribed such that $\mathbf{x}_N$ is approximately distributed according to $\mathcal{N}(\mathbf{0}, \mathbf{I})$. A variational Markov chain in the reverse direction is parameterized with $p_{\boldsymbol{\theta}}(\mathbf{x}_{i-1} | \mathbf{x}_i) = \mathcal{N}(\mathbf{x}_{i-1}; \frac{1}{\sqrt{1 - \beta_i}}(\mathbf{x}_i + \beta_i \mathbf{s}_{\boldsymbol{\theta}}(\mathbf{x}_i, i)), \beta_i \mathbf{I})$, and trained with a re-weighted variant of the evidence lower bound (ELBO):

$$\boldsymbol{\theta}^* = \arg\min_{\boldsymbol{\theta}} \sum_{i=1}^{N} (1 - \alpha_i) \mathbb{E}_{p_{\text{data}}(\mathbf{x})} \mathbb{E}_{p_{\alpha_i}(\tilde{\mathbf{x}}|\mathbf{x})} [\|\mathbf{s}_{\boldsymbol{\theta}}(\tilde{\mathbf{x}}, i) - \nabla_{\tilde{\mathbf{x}}} \log p_{\alpha_i}(\tilde{\mathbf{x}} \mid \mathbf{x}) \|_2^2]. \tag{3}$$

After solving Eq. (3) to get the optimal model $\mathbf{s}_{\boldsymbol{\theta}^*}(\mathbf{x}, i)$, samples can be generated by starting from $\mathbf{x}_N \sim \mathcal{N}(\mathbf{0}, \mathbf{I})$ and following the estimated reverse Markov chain as below

$$\mathbf{x}_{i-1} = \frac{1}{\sqrt{1 - \beta_i}}(\mathbf{x}_i + \beta_i \mathbf{s}_{\boldsymbol{\theta}^*}(\mathbf{x}_i, i)) + \sqrt{\beta_i} \mathbf{z}_i, \quad i = N, N - 1, \cdots, 1. \tag{4}$$

We call this method *ancestral sampling*, since it amounts to performing ancestral sampling from the graphical model $\prod_{i=1}^{N} p_{\boldsymbol{\theta}}(\mathbf{x}_{i-1} \mid \mathbf{x}_i)$. The objective Eq. (3) described here is $L_{\text{simple}}$ in Ho et al. (2020), written in a form to expose more similarity to Eq. (1). Like Eq. (1), Eq. (3) is also a weighted sum of denoising score matching objectives, which implies that the optimal model, $\mathbf{s}_{\boldsymbol{\theta}^*}(\tilde{\mathbf{x}}, i)$, matches the score of the perturbed data distribution, $\nabla_{\mathbf{x}} \log p_{\alpha_i}(\mathbf{x})$. Notably, the weights of the $i$-th summand in Eq. (1) and Eq. (3), namely $\sigma_i^2$ and $(1 - \alpha_i)$, are related to corresponding perturbation kernels in the same functional form: $\sigma_i^2 \propto 1/\mathbb{E}[\|\nabla_{\mathbf{x}} \log p_{\sigma_i}(\tilde{\mathbf{x}} \mid \mathbf{x})\|_2^2]$ and $(1 - \alpha_i) \propto 1/\mathbb{E}[\|\nabla_{\mathbf{x}} \log p_{\alpha_i}(\tilde{\mathbf{x}} \mid \mathbf{x})\|_2^2]$.

## 3 SCORE-BASED GENERATIVE MODELING WITH SDEs

Perturbing data with multiple noise scales is key to the success of previous methods. We propose to generalize this idea further to an infinite number of noise scales, such that perturbed data distributions evolve according to an SDE as the noise intensifies. An overview of our framework is given in Fig. 2.

### 3.1 PERTURBING DATA WITH SDEs

Our goal is to construct a diffusion process $\{\mathbf{x}(t)\}_{t=0}^{T}$ indexed by a continuous time variable $t \in [0, T]$, such that $\mathbf{x}(0) \sim p_0$, for which we have a dataset of i.i.d. samples, and $\mathbf{x}(T) \sim p_T$, for which we have a tractable form to generate samples efficiently. In other words, $p_0$ is the data distribution and $p_T$ is the prior distribution. This diffusion process can be modeled as the solution to an Itô SDE:

$$\mathrm{d}\mathbf{x} = \mathbf{f}(\mathbf{x}, t)\mathrm{d}t + g(t)\mathrm{d}\mathbf{w}, \tag{5}$$

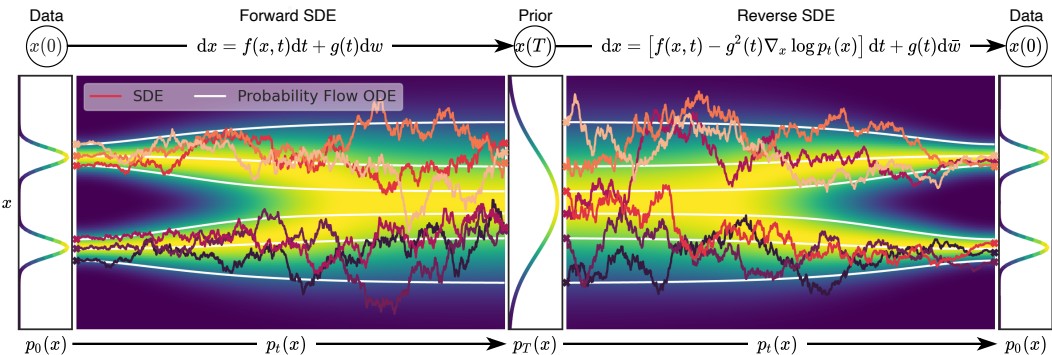

Figure 2: **Overview of score-based generative modeling through SDEs**. We can map data to a noise distribution (the prior) with an SDE (Section 3.1), and reverse this SDE for generative modeling (Section 3.2). We can also reverse the associated probability flow ODE (Section 4.3), which yields a deterministic process that samples from the same distribution as the SDE. Both the reverse-time SDE and probability flow ODE can be obtained by estimating the score $\nabla_{\mathbf{x}} \log p_t(\mathbf{x})$ (Section 3.3).

where $\mathbf{w}$ is the standard Wiener process (a.k.a., Brownian motion), $\mathbf{f}(\cdot, t) : \mathbb{R}^d \to \mathbb{R}^d$ is a vector-valued function called the *drift* coefficient of $\mathbf{x}(t)$, and $g(\cdot) : \mathbb{R} \to \mathbb{R}$ is a scalar function known as the *diffusion* coefficient of $\mathbf{x}(t)$. For ease of presentation we assume the diffusion coefficient is a scalar (instead of a $d \times d$ matrix) and does not depend on $\mathbf{x}$, but our theory can be generalized to hold in those cases (see Appendix A). The SDE has a unique strong solution as long as the coefficients are globally Lipschitz in both state and time (Øksendal, 2003). We hereafter denote by $p_t(\mathbf{x})$ the probability density of $\mathbf{x}(t)$, and use $p_{st}(\mathbf{x}(t) \mid \mathbf{x}(s))$ to denote the transition kernel from $\mathbf{x}(s)$ to $\mathbf{x}(t)$, where $0 \leqslant s < t \leqslant T$.

Typically, $p_T$ is an unstructured prior distribution that contains no information of $p_0$, such as a Gaussian distribution with fixed mean and variance. There are various ways of designing the SDE in Eq. (5) such that it diffuses the data distribution into a fixed prior distribution. We provide several examples later in Section 3.4 that are derived from continuous generalizations of SMLD and DDPM.

## 3.2 Generating samples by reversing the SDE

By starting from samples of $\mathbf{x}(T) \sim p_T$ and reversing the process, we can obtain samples $\mathbf{x}(0) \sim p_0$. A remarkable result from Anderson (1982) states that the reverse of a diffusion process is also a diffusion process, running backwards in time and given by the reverse-time SDE:

$$d\mathbf{x} = [\mathbf{f}(\mathbf{x}, t) - g(t)^2 \nabla_{\mathbf{x}} \log p_t(\mathbf{x})]dt + g(t)d\bar{\mathbf{w}}, \tag{6}$$

where $\bar{\mathbf{w}}$ is a standard Wiener process when time flows backwards from $T$ to $0$, and $dt$ is an infinitesimal negative timestep. Once the score of each marginal distribution, $\nabla_{\mathbf{x}} \log p_t(\mathbf{x})$, is known for all $t$, we can derive the reverse diffusion process from Eq. (6) and simulate it to sample from $p_0$.

## 3.3 Estimating scores for the SDE

The score of a distribution can be estimated by training a score-based model on samples with score matching (Hyvärinen, 2005; Song et al., 2019a). To estimate $\nabla_{\mathbf{x}} \log p_t(\mathbf{x})$, we can train a time-dependent score-based model $\mathbf{s}_{\boldsymbol{\theta}}(\mathbf{x}, t)$ via a continuous generalization to Eqs. (1) and (3):

$$\boldsymbol{\theta}^* = \arg\min_{\boldsymbol{\theta}} \mathbb{E}_t \left\{ \lambda(t) \mathbb{E}_{\mathbf{x}(0)} \mathbb{E}_{\mathbf{x}(t)|\mathbf{x}(0)} \left[ \left\| \mathbf{s}_{\boldsymbol{\theta}}(\mathbf{x}(t), t) - \nabla_{\mathbf{x}(t)} \log p_{0t}(\mathbf{x}(t) \mid \mathbf{x}(0)) \right\|_2^2 \right] \right\}. \tag{7}$$

Here $\lambda : [0, T] \to \mathbb{R}_{>0}$ is a positive weighting function, $t$ is uniformly sampled over $[0, T]$, $\mathbf{x}(0) \sim p_0(\mathbf{x})$ and $\mathbf{x}(t) \sim p_{0t}(\mathbf{x}(t) \mid \mathbf{x}(0))$. With sufficient data and model capacity, score matching ensures that the optimal solution to Eq. (7), denoted by $\mathbf{s}_{\boldsymbol{\theta}*}(\mathbf{x}, t)$, equals $\nabla_{\mathbf{x}} \log p_t(\mathbf{x})$ for almost all $\mathbf{x}$ and $t$. As in SMLD and DDPM, we can typically choose $\lambda \propto 1/\mathbb{E} \left[ \left\| \nabla_{\mathbf{x}(t)} \log p_{0t}(\mathbf{x}(t) \mid \mathbf{x}(0)) \right\|_2^2 \right]$. Note that Eq. (7) uses denoising score matching, but other score matching objectives, such as sliced

score matching (Song et al., 2019a) and finite-difference score matching (Pang et al., 2020) are also applicable here.

We typically need to know the transition kernel $p_{0t}(\mathbf{x}(t) \mid \mathbf{x}(0))$ to efficiently solve Eq. (7). When $\mathbf{f}(\cdot, t)$ is affine, the transition kernel is always a Gaussian distribution, where the mean and variance are often known in closed-forms and can be obtained with standard techniques (see Section 5.5 in Särkkä & Solin (2019)). For more general SDEs, we may solve Kolmogorov's forward equation (Øksendal, 2003) to obtain $p_{0t}(\mathbf{x}(t) \mid \mathbf{x}(0))$. Alternatively, we can simulate the SDE to sample from $p_{0t}(\mathbf{x}(t) \mid \mathbf{x}(0))$ and replace denoising score matching in Eq. (7) with sliced score matching for model training, which bypasses the computation of $\nabla_{\mathbf{x}(t)} \log p_{0t}(\mathbf{x}(t) \mid \mathbf{x}(0))$ (see Appendix A).

### 3.4 Examples: VE, VP SDEs and beyond

The noise perturbations used in SMLD and DDPM can be regarded as discretizations of two different SDEs. Below we provide a brief discussion and relegate more details to Appendix B.

When using a total of $N$ noise scales, each perturbation kernel $p_{\sigma_i}(\mathbf{x} \mid \mathbf{x}_0)$ of SMLD corresponds to the distribution of $\mathbf{x}_i$ in the following Markov chain:

$$\mathbf{x}_i = \mathbf{x}_{i-1} + \sqrt{\sigma_i^2 - \sigma_{i-1}^2}\mathbf{z}_{i-1}, \quad i = 1, \cdots, N, \tag{8}$$

where $\mathbf{z}_{i-1} \sim \mathcal{N}(\mathbf{0}, \mathbf{I})$, and we have introduced $\sigma_0 = 0$ to simplify the notation. In the limit of $N \to \infty$, $\{\sigma_i\}_{i=1}^N$ becomes a function $\sigma(t)$, $\mathbf{z}_i$ becomes $\mathbf{z}(t)$, and the Markov chain $\{\mathbf{x}_i\}_{i=1}^N$ becomes a continuous stochastic process $\{\mathbf{x}(t)\}_{t=0}^1$, where we have used a continuous time variable $t \in [0, 1]$ for indexing, rather than an integer $i$. The process $\{\mathbf{x}(t)\}_{t=0}^1$ is given by the following SDE

$$\mathrm{d}\mathbf{x} = \sqrt{\frac{\mathrm{d}\left[\sigma^2(t)\right]}{\mathrm{d}t}}\mathrm{d}\mathbf{w}. \tag{9}$$

Likewise for the perturbation kernels $\{p_{\alpha_i}(\mathbf{x} \mid \mathbf{x}_0)\}_{i=1}^N$ of DDPM, the discrete Markov chain is

$$\mathbf{x}_i = \sqrt{1 - \beta_i}\mathbf{x}_{i-1} + \sqrt{\beta_i}\mathbf{z}_{i-1}, \quad i = 1, \cdots, N. \tag{10}$$

As $N \to \infty$, Eq. (10) converges to the following SDE,

$$\mathrm{d}\mathbf{x} = -\frac{1}{2}\beta(t)\mathbf{x}\,\mathrm{d}t + \sqrt{\beta(t)}\,\mathrm{d}\mathbf{w}. \tag{11}$$

Therefore, the noise perturbations used in SMLD and DDPM correspond to discretizations of SDEs Eqs. (9) and (11). Interestingly, the SDE of Eq. (9) always gives a process with exploding variance when $t \to \infty$, whilst the SDE of Eq. (11) yields a process with a fixed variance of one when the initial distribution has unit variance (proof in Appendix B). Due to this difference, we hereafter refer to Eq. (9) as the Variance Exploding (VE) SDE, and Eq. (11) the Variance Preserving (VP) SDE.

Inspired by the VP SDE, we propose a new type of SDEs which perform particularly well on likelihoods (see Section 4.3), given by

$$\mathrm{d}\mathbf{x} = -\frac{1}{2}\beta(t)\mathbf{x}\,\mathrm{d}t + \sqrt{\beta(t)(1 - e^{-2\int_0^t \beta(s)\mathrm{d}s})}\mathrm{d}\mathbf{w}. \tag{12}$$

When using the same $\beta(t)$ and starting from the same initial distribution, the variance of the stochastic process induced by Eq. (12) is always bounded by the VP SDE at every intermediate time step (proof in Appendix B). For this reason, we name Eq. (12) the sub-VP SDE.

Since VE, VP and sub-VP SDEs all have affine drift coefficients, their perturbation kernels $p_{0t}(\mathbf{x}(t) \mid \mathbf{x}(0))$ are all Gaussian and can be computed in closed-forms, as discussed in Section 3.3. This makes training with Eq. (7) particularly efficient.

## 4 Solving the reverse SDE

After training a time-dependent score-based model $\mathbf{s}_{\boldsymbol{\theta}}$, we can use it to construct the reverse-time SDE and then simulate it with numerical approaches to generate samples from $p_0$.

Table 1: Comparing different reverse-time SDE solvers on CIFAR-10. Shaded regions are obtained with the same computation (number of score function evaluations). Mean and standard deviation are reported over five sampling runs. "P1000" or "P2000": predictor-only samplers using 1000 or 2000 steps. "C2000": corrector-only samplers using 2000 steps. "PC1000": Predictor-Corrector (PC) samplers using 1000 predictor and 1000 corrector steps.

| FID↓ \ Sampler  Predictor | Variance Exploding SDE (SMLD) | | | | Variance Preserving SDE (DDPM) | | | |
|---|---|---|---|---|---|---|---|---|
| | P1000 | P2000 | C2000 | PC1000 | P1000 | P2000 | C2000 | PC1000 |
| ancestral sampling | $4.98_{\pm .06}$ | $4.88_{\pm .06}$ | | $\mathbf{3.62}_{\pm \mathbf{.03}}$ | $3.24_{\pm .02}$ | $3.24_{\pm .02}$ | | $\mathbf{3.21}_{\pm \mathbf{.02}}$ |
| reverse diffusion | $4.79_{\pm .07}$ | $4.74_{\pm .08}$ | $20.43_{\pm .07}$ | $\mathbf{3.60}_{\pm \mathbf{.02}}$ | $3.21_{\pm .02}$ | $3.19_{\pm .02}$ | $19.06_{\pm .06}$ | $\mathbf{3.18}_{\pm \mathbf{.01}}$ |
| probability flow | $15.41_{\pm .15}$ | $10.54_{\pm .08}$ | | $\mathbf{3.51}_{\pm \mathbf{.04}}$ | $3.59_{\pm .04}$ | $3.23_{\pm .03}$ | | $\mathbf{3.06}_{\pm \mathbf{.03}}$ |

## 4.1 GENERAL-PURPOSE NUMERICAL SDE SOLVERS

Numerical solvers provide approximate trajectories from SDEs. Many general-purpose numerical methods exist for solving SDEs, such as Euler-Maruyama and stochastic Runge-Kutta methods (Kloeden & Platen, 2013), which correspond to different discretizations of the stochastic dynamics. We can apply any of them to the reverse-time SDE for sample generation.

Ancestral sampling, the sampling method of DDPM (Eq. (4)), actually corresponds to one special discretization of the reverse-time VP SDE (Eq. (11)) (see Appendix E). Deriving the ancestral sampling rules for new SDEs, however, can be non-trivial. To remedy this, we propose *reverse diffusion samplers* (details in Appendix E), which discretize the reverse-time SDE in the same way as the forward one, and thus can be readily derived given the forward discretization. As shown in Table 1, reverse diffusion samplers perform slightly better than ancestral sampling for both SMLD and DDPM models on CIFAR-10 (DDPM-type ancestral sampling is also applicable to SMLD models, see Appendix F.)

## 4.2 PREDICTOR-CORRECTOR SAMPLERS

Unlike generic SDEs, we have additional information that can be used to improve solutions. Since we have a score-based model $\mathbf{s}_{\boldsymbol{\theta}*}(\mathbf{x}, t) \approx \nabla_{\mathbf{x}} \log p_t(\mathbf{x})$, we can employ score-based MCMC approaches, such as Langevin MCMC (Parisi, 1981; Grenander & Miller, 1994) or HMC (Neal et al., 2011) to sample from $p_t$ directly, and correct the solution of a numerical SDE solver.

Specifically, at each time step, the numerical SDE solver first gives an estimate of the sample at the next time step, playing the role of a "predictor". Then, the score-based MCMC approach corrects the marginal distribution of the estimated sample, playing the role of a "corrector". The idea is analogous to Predictor-Corrector methods, a family of numerical continuation techniques for solving systems of equations (Allgower & Georg, 2012), and we similarly name our hybrid sampling algorithms *Predictor-Corrector* (PC) samplers. Please find pseudo-code and a complete description in Appendix G. PC samplers generalize the original sampling methods of SMLD and DDPM: the former uses an identity function as the predictor and annealed Langevin dynamics as the corrector, while the latter uses ancestral sampling as the predictor and identity as the corrector.

We test PC samplers on SMLD and DDPM models (see Algorithms 2 and 3 in Appendix G) trained with original discrete objectives given by Eqs. (1) and (3). This exhibits the compatibility of PC samplers to score-based models trained with a fixed number of noise scales. We summarize the performance of different samplers in Table 1, where probability flow is a predictor to be discussed in Section 4.3. Detailed experimental settings and additional results are given in Appendix G. We observe that our reverse diffusion sampler always outperform ancestral sampling, and corrector-only methods (C2000) perform worse than other competitors (P2000, PC1000) with the same computation (In fact, we need way more corrector steps per noise scale, and thus more computation, to match the performance of other samplers.) For all predictors, adding one corrector step for each predictor step (PC1000) doubles computation but always improves sample quality (against P1000). Moreover, it is typically better than doubling the number of predictor steps without adding a corrector (P2000), where we have to interpolate between noise scales in an ad hoc manner (detailed in Appendix G) for SMLD/DDPM models. In Fig. 9 (Appendix G), we additionally provide qualitative comparison for

Table 2: NLLs and FIDs (ODE) on CIFAR-10.

| Model | NLL Test ↓ | FID ↓ |
|---|---|---|
| RealNVP (Dinh et al., 2016) | 3.49 | - |
| iResNet (Behrmann et al., 2019) | 3.45 | - |
| Glow (Kingma & Dhariwal, 2018) | 3.35 | - |
| MintNet (Song et al., 2019b) | 3.32 | - |
| Residual Flow (Chen et al., 2019) | 3.28 | 46.37 |
| FFJORD (Grathwohl et al., 2018) | 3.40 | - |
| Flow++ (Ho et al., 2019) | 3.29 | - |
| DDPM ($L$) (Ho et al., 2020) | $\leqslant 3.70^*$ | 13.51 |
| DDPM ($L_{\text{simple}}$) (Ho et al., 2020) | $\leqslant 3.75^*$ | 3.17 |
| DDPM | 3.28 | 3.37 |
| DDPM cont. (VP) | 3.21 | 3.69 |
| DDPM cont. (sub-VP) | 3.05 | 3.56 |
| DDPM++ cont. (VP) | 3.16 | 3.93 |
| DDPM++ cont. (sub-VP) | 3.02 | 3.16 |
| DDPM++ cont. (deep, VP) | 3.13 | 3.08 |
| DDPM++ cont. (deep, sub-VP) | **2.99** | **2.92** |

Table 3: CIFAR-10 sample quality.

| Model | FID↓ | IS↑ |
|---|---|---|
| **Conditional** | | |
| BigGAN (Brock et al., 2018) | 14.73 | 9.22 |
| StyleGAN2-ADA (Karras et al., 2020a) | **2.42** | **10.14** |
| **Unconditional** | | |
| StyleGAN2-ADA (Karras et al., 2020a) | 2.92 | 9.83 |
| NCSN (Song & Ermon, 2019) | 25.32 | 8.87 ± .12 |
| NCSNv2 (Song & Ermon, 2020) | 10.87 | 8.40 ± .07 |
| DDPM (Ho et al., 2020) | 3.17 | 9.46 ± .11 |
| DDPM++ | 2.78 | 9.64 |
| DDPM++ cont. (VP) | 2.55 | 9.58 |
| DDPM++ cont. (sub-VP) | 2.61 | 9.56 |
| DDPM++ cont. (deep, VP) | 2.41 | 9.68 |
| DDPM++ cont. (deep, sub-VP) | 2.41 | 9.57 |
| NCSN++ | 2.45 | 9.73 |
| NCSN++ cont. (VE) | 2.38 | 9.83 |
| NCSN++ cont. (deep, VE) | **2.20** | **9.89** |

models trained with the continuous objective Eq. (7) on $256 \times 256$ LSUN images and the VE SDE, where PC samplers clearly surpass predictor-only samplers under comparable computation, when using a proper number of corrector steps.

### 4.3 PROBABILITY FLOW AND CONNECTION TO NEURAL ODEs

Score-based models enable another numerical method for solving the reverse-time SDE. For all diffusion processes, there exists a corresponding *deterministic process* whose trajectories share the same marginal probability densities $\{p_t(\mathbf{x})\}_{t=0}^{T}$ as the SDE. This deterministic process satisfies an ODE (more details in Appendix D.1):

$$\mathrm{d}\mathbf{x} = \Big[\mathbf{f}(\mathbf{x},t) - \frac{1}{2}g(t)^2 \nabla_{\mathbf{x}} \log p_t(\mathbf{x})\Big]\mathrm{d}t, \qquad (13)$$

which can be determined from the SDE once scores are known. We name the ODE in Eq. (13) the *probability flow ODE*. When the score function is approximated by the time-dependent score-based model, which is typically a neural network, this is an example of a neural ODE (Chen et al., 2018).

**Exact likelihood computation** Leveraging the connection to neural ODEs, we can compute the density defined by Eq. (13) via the instantaneous change of variables formula (Chen et al., 2018). This allows us to compute the *exact likelihood on any input data* (details in Appendix D.2). As an example, we report negative log-likelihoods (NLLs) measured in bits/dim on the CIFAR-10 dataset in Table 2. We compute log-likelihoods on uniformly dequantized data, and only compare to models evaluated in the same way (omitting models evaluated with variational dequantization (Ho et al., 2019) or discrete data), except for DDPM ($L/L_{\text{simple}}$) whose ELBO values (annotated with *) are reported on discrete data. Main results: (i) For the same DDPM model in Ho et al. (2020), we obtain better bits/dim than ELBO, since our likelihoods are exact; (ii) Using the same architecture, we trained another DDPM model with the continuous objective in Eq. (7) (*i.e.*, DDPM cont.), which further improves the likelihood; (iii) With sub-VP SDEs, we always get higher likelihoods compared to VP SDEs; (iv) With improved architecture (*i.e.*, DDPM++ cont., details in Section 4.4) and the sub-VP SDE, we can set a new record bits/dim of 2.99 on uniformly dequantized CIFAR-10 even *without maximum likelihood training*.

**Manipulating latent representations** By integrating Eq. (13), we can encode any datapoint $\mathbf{x}(0)$ into a latent space $\mathbf{x}(T)$. Decoding can be achieved by integrating a corresponding ODE for the reverse-time SDE. As is done with other invertible models such as neural ODEs and normalizing flows (Dinh et al., 2016; Kingma & Dhariwal, 2018), we can manipulate this latent representation for image editing, such as interpolation, and temperature scaling (see Fig. 3 and Appendix D.4).

**Uniquely identifiable encoding** Unlike most current invertible models, our encoding is *uniquely identifiable*, meaning that with sufficient training data, model capacity, and optimization accuracy, the encoding for an input is uniquely determined by the data distribution (Roeder et al., 2020). This is because our forward SDE, Eq. (5), has no trainable parameters, and its associated probability flow

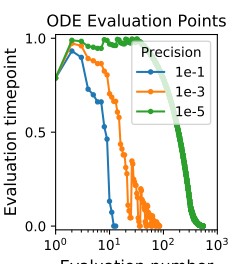 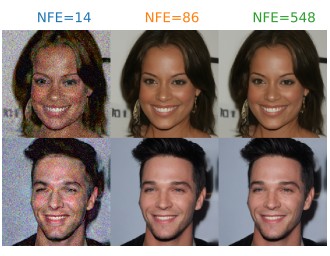 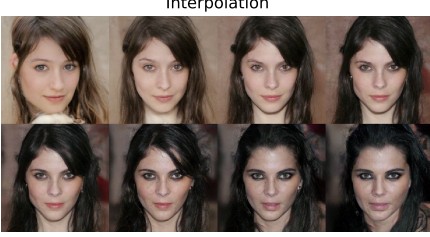

Figure 3: **Probability flow ODE enables fast sampling** with adaptive stepsizes as the numerical precision is varied (*left*), and reduces the number of score function evaluations (NFE) without harming quality (*middle*). The invertible mapping from latents to images allows for interpolations (*right*).

ODE, Eq. (13), provides the same trajectories given perfectly estimated scores. We provide additional empirical verification on this property in Appendix D.5.

**Efficient sampling** As with neural ODEs, we can sample $\mathbf{x}(0) \sim p_0$ by solving Eq. (13) from different final conditions $\mathbf{x}(T) \sim p_T$. Using a fixed discretization strategy we can generate competitive samples, especially when used in conjuction with correctors (Table 1, "probability flow sampler", details in Appendix D.3). Using a black-box ODE solver (Dormand & Prince, 1980) not only produces high quality samples (Table 2, details in Appendix D.4), but also allows us to explicitly trade-off accuracy for efficiency. With a larger error tolerance, the number of function evaluations can be reduced by over $90\%$ without affecting the visual quality of samples (Fig. 3).

### 4.4 ARCHITECTURE IMPROVEMENTS

We explore several new architecture designs for score-based models using both VE and VP SDEs (details in Appendix H), where we train models with the same discrete objectives as in SMLD/DDPM. We directly transfer the architectures for VP SDEs to sub-VP SDEs due to their similarity. Our optimal architecture for the VE SDE, named NCSN++, achieves an FID of 2.45 on CIFAR-10 with PC samplers, while our optimal architecture for the VP SDE, called DDPM++, achieves 2.78.

By switching to the continuous training objective in Eq. (7), and increasing the network depth, we can further improve sample quality for all models. The resulting architectures are denoted as NCSN++ cont. and DDPM++ cont. in Table 3 for VE and VP/sub-VP SDEs respectively. Results reported in Table 3 are for the checkpoint with the smallest FID over the course of training, where samples are generated with PC samplers. In contrast, FID scores and NLL values in Table 2 are reported for the last training checkpoint, and samples are obtained with black-box ODE solvers. As shown in Table 3, VE SDEs typically provide better sample quality than VP/sub-VP SDEs, but we also empirically observe that their likelihoods are worse than VP/sub-VP SDE counterparts. This indicates that practitioners likely need to experiment with different SDEs for varying domains and architectures.

Our best model for sample quality, NCSN++ cont. (deep, VE), doubles the network depth and sets new records for both inception score and FID on unconditional generation for CIFAR-10. Surprisingly, we can achieve better FID than the previous best conditional generative model without requiring labeled data. With all improvements together, we also obtain the first set of high-fidelity samples on CelebA-HQ $1024 \times 1024$ from score-based models (see Appendix H.3). Our best model for likelihoods, DDPM++ cont. (deep, sub-VP), similarly doubles the network depth and achieves a log-likelihood of 2.99 bits/dim with the continuous objective in Eq. (7). To our best knowledge, this is the highest likelihood on uniformly dequantized CIFAR-10.

## 5 CONTROLLABLE GENERATION

The continuous structure of our framework allows us to not only produce data samples from $p_0$, but also from $p_0(\mathbf{x}(0) \mid \mathbf{y})$ if $p_t(\mathbf{y} \mid \mathbf{x}(t))$ is known. Given a forward SDE as in Eq. (5), we can sample

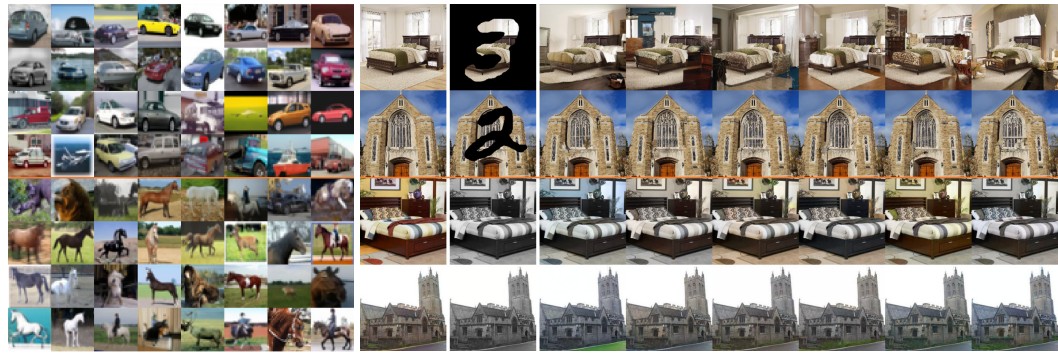

Figure 4: *Left*: Class-conditional samples on $32 \times 32$ CIFAR-10. Top four rows are automobiles and bottom four rows are horses. *Right:* Inpainting (top two rows) and colorization (bottom two rows) results on $256 \times 256$ LSUN. First column is the original image, second column is the masked/grayscale image, remaining columns are sampled image completions or colorizations.

from $p_t(\mathbf{x}(t) \mid \mathbf{y})$ by starting from $p_T(\mathbf{x}(T) \mid \mathbf{y})$ and solving a conditional reverse-time SDE:

$$d\mathbf{x} = \{\mathbf{f}(\mathbf{x}, t) - g(t)^2[\nabla_{\mathbf{x}} \log p_t(\mathbf{x}) + \nabla_{\mathbf{x}} \log p_t(\mathbf{y} \mid \mathbf{x})]\}dt + g(t)d\bar{\mathbf{w}}. \qquad (14)$$

In general, we can use Eq. (14) to solve a large family of *inverse problems* with score-based generative models, once given an estimate of the gradient of the forward process, $\nabla_{\mathbf{x}} \log p_t(\mathbf{y} \mid \mathbf{x}(t))$. In some cases, it is possible to train a separate model to learn the forward process $\log p_t(\mathbf{y} \mid \mathbf{x}(t))$ and compute its gradient. Otherwise, we may estimate the gradient with heuristics and domain knowledge. In Appendix I.4, we provide a broadly applicable method for obtaining such an estimate without the need of training auxiliary models.

We consider three applications of controllable generation with this approach: class-conditional generation, image imputation and colorization. When $\mathbf{y}$ represents class labels, we can train a time-dependent classifier $p_t(\mathbf{y} \mid \mathbf{x}(t))$ for class-conditional sampling. Since the forward SDE is tractable, we can easily create training data $(\mathbf{x}(t), \mathbf{y})$ for the time-dependent classifier by first sampling $(\mathbf{x}(0), \mathbf{y})$ from a dataset, and then sampling $\mathbf{x}(t) \sim p_{0t}(\mathbf{x}(t) \mid \mathbf{x}(0))$. Afterwards, we may employ a mixture of cross-entropy losses over different time steps, like Eq. (7), to train the time-dependent classifier $p_t(\mathbf{y} \mid \mathbf{x}(t))$. We provide class-conditional CIFAR-10 samples in Fig. 4 (left), and relegate more details and results to Appendix I.

Imputation is a special case of conditional sampling. Suppose we have an incomplete data point $\mathbf{y}$ where only some subset, $\Omega(\mathbf{y})$ is known. Imputation amounts to sampling from $p(\mathbf{x}(0) \mid \Omega(\mathbf{y}))$, which we can accomplish using an unconditional model (see Appendix I.2). Colorization is a special case of imputation, except that the known data dimensions are coupled. We can decouple these data dimensions with an orthogonal linear transformation, and perform imputation in the transformed space (details in Appendix I.3). Fig. 4 (right) shows results for inpainting and colorization achieved with unconditional time-dependent score-based models.

## 6 CONCLUSION

We presented a framework for score-based generative modeling based on SDEs. Our work enables a better understanding of existing approaches, new sampling algorithms, exact likelihood computation, uniquely identifiable encoding, latent code manipulation, and brings new conditional generation abilities to the family of score-based generative models.

While our proposed sampling approaches improve results and enable more efficient sampling, they remain slower at sampling than GANs (Goodfellow et al., 2014) on the same datasets. Identifying ways of combining the stable learning of score-based generative models with the fast sampling of implicit models like GANs remains an important research direction. Additionally, the breadth of samplers one can use when given access to score functions introduces a number of hyper-parameters. Future work would benefit from improved methods to automatically select and tune these hyper-parameters, as well as more extensive investigation on the merits and limitations of various samplers.

ACKNOWLEDGEMENTS

We would like to thank Nanxin Chen, Ruiqi Gao, Jonathan Ho, Kevin Murphy, Tim Salimans and Han Zhang for their insightful discussions during the course of this project. This research was partially supported by NSF (#1651565, #1522054, #1733686), ONR (N000141912145), AFOSR (FA95501910024), and TensorFlow Research Cloud. Yang Song was partially supported by the Apple PhD Fellowship in AI/ML.

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

APPENDIX

We include several appendices with additional details, derivations, and results. Our framework allows general SDEs with matrix-valued diffusion coefficients that depend on the state, for which we provide a detailed discussion in Appendix A. We give a full derivation of VE, VP and sub-VP SDEs in Appendix B, and discuss how to use them from a practitioner's perspective in Appendix C. We elaborate on the probability flow formulation of our framework in Appendix D, including a derivation of the probability flow ODE (Appendix D.1), exact likelihood computation (Appendix D.2), probability flow sampling with a fixed discretization strategy (Appendix D.3), sampling with black-box ODE solvers (Appendix D.4), and experimental verification on uniquely identifiable encoding (Appendix D.5). We give a full description of the reverse diffusion sampler in Appendix E, the DDPM-type ancestral sampler for SMLD models in Appendix F, and Predictor-Corrector samplers in Appendix G. We explain our model architectures and detailed experimental settings in Appendix H, with $1024 \times 1024$ CelebA-HQ samples therein. Finally, we detail on the algorithms for controllable generation in Appendix I, and include extended results for class-conditional generation (Appendix I.1), image inpainting (Appendix I.2), colorization (Appendix I.3), and a strategy for solving general inverse problems (Appendix I.4).

## A    THE FRAMEWORK FOR MORE GENERAL SDEs

In the main text, we introduced our framework based on a simplified SDE Eq. (5) where the diffusion coefficient is independent of $\mathbf{x}(t)$. It turns out that our framework can be extended to hold for more general diffusion coefficients. We can consider SDEs in the following form:

$$\mathrm{d}\mathbf{x} = \mathbf{f}(\mathbf{x}, t)\mathrm{d}t + \mathbf{G}(\mathbf{x}, t)\mathrm{d}\mathbf{w}, \tag{15}$$

where $\mathbf{f}(\cdot, t) : \mathbb{R}^d \to \mathbb{R}^d$ and $\mathbf{G}(\cdot, t) : \mathbb{R}^d \to \mathbb{R}^{d \times d}$. We follow the Itô interpretation of SDEs throughout this paper.

According to (Anderson, 1982), the reverse-time SDE is given by (cf., Eq. (6))

$$\mathrm{d}\mathbf{x} = \{\mathbf{f}(\mathbf{x}, t) - \nabla \cdot [\mathbf{G}(\mathbf{x}, t)\mathbf{G}(\mathbf{x}, t)^\mathsf{T}] - \mathbf{G}(\mathbf{x}, t)\mathbf{G}(\mathbf{x}, t)^\mathsf{T}\nabla_\mathbf{x} \log p_t(\mathbf{x})\}\mathrm{d}t + \mathbf{G}(\mathbf{x}, t)\mathrm{d}\bar{\mathbf{w}}, \tag{16}$$

where we define $\nabla \cdot \mathbf{F}(\mathbf{x}) := (\nabla \cdot \mathbf{f}^1(\mathbf{x}), \nabla \cdot \mathbf{f}^2(\mathbf{x}), \cdots, \nabla \cdot \mathbf{f}^d(\mathbf{x}))^\mathsf{T}$ for a matrix-valued function $\mathbf{F}(\mathbf{x}) := (\mathbf{f}^1(\mathbf{x}), \mathbf{f}^2(\mathbf{x}), \cdots, \mathbf{f}^d(\mathbf{x}))^\mathsf{T}$ throughout the paper.

The probability flow ODE corresponding to Eq. (15) has the following form (cf., Eq. (13), see a detailed derivation in Appendix D.1):

$$\mathrm{d}\mathbf{x} = \left\{\mathbf{f}(\mathbf{x}, t) - \frac{1}{2}\nabla \cdot [\mathbf{G}(\mathbf{x}, t)\mathbf{G}(\mathbf{x}, t)^\mathsf{T}] - \frac{1}{2}\mathbf{G}(\mathbf{x}, t)\mathbf{G}(\mathbf{x}, t)^\mathsf{T}\nabla_\mathbf{x} \log p_t(\mathbf{x})\right\}\mathrm{d}t. \tag{17}$$

Finally for conditional generation with the general SDE Eq. (15), we can solve the conditional reverse-time SDE below (cf., Eq. (14), details in Appendix I):

$$\begin{aligned}\mathrm{d}\mathbf{x} = \{\mathbf{f}(\mathbf{x}, t) &- \nabla \cdot [\mathbf{G}(\mathbf{x}, t)\mathbf{G}(\mathbf{x}, t)^\mathsf{T}] - \mathbf{G}(\mathbf{x}, t)\mathbf{G}(\mathbf{x}, t)^\mathsf{T}\nabla_\mathbf{x} \log p_t(\mathbf{x}) \\ &- \mathbf{G}(\mathbf{x}, t)\mathbf{G}(\mathbf{x}, t)^\mathsf{T}\nabla_\mathbf{x} \log p_t(\mathbf{y} \mid \mathbf{x})\}\mathrm{d}t + \mathbf{G}(\mathbf{x}, t)\mathrm{d}\bar{\mathbf{w}}. \end{aligned} \tag{18}$$

When the drift and diffusion coefficient of an SDE are not affine, it can be difficult to compute the transition kernel $p_{0t}(\mathbf{x}(t) \mid \mathbf{x}(0))$ in closed form. This hinders the training of score-based models, because Eq. (7) requires knowing $\nabla_{\mathbf{x}(t)} \log p_{0t}(\mathbf{x}(t) \mid \mathbf{x}(0))$. To overcome this difficulty, we can replace denoising score matching in Eq. (7) with other efficient variants of score matching that do not require computing $\nabla_{\mathbf{x}(t)} \log p_{0t}(\mathbf{x}(t) \mid \mathbf{x}(0))$. For example, when using sliced score matching (Song et al., 2019a), our training objective Eq. (7) becomes

$$\boldsymbol{\theta}^* = \arg\min_{\boldsymbol{\theta}} \mathbb{E}_t \left\{\lambda(t)\mathbb{E}_{\mathbf{x}(0)}\mathbb{E}_{\mathbf{x}(t)}\mathbb{E}_{\mathbf{v} \sim p_\mathbf{v}}\left[\frac{1}{2}\|\mathbf{s}_{\boldsymbol{\theta}}(\mathbf{x}(t), t)\|_2^2 + \mathbf{v}^\mathsf{T}\mathbf{s}_{\boldsymbol{\theta}}(\mathbf{x}(t), t)\mathbf{v}\right]\right\}, \tag{19}$$

where $\lambda : [0, T] \to \mathbb{R}^+$ is a positive weighting function, $t \sim \mathcal{U}(0, T)$, $\mathbb{E}[\mathbf{v}] = \mathbf{0}$, and $\mathrm{Cov}[\mathbf{v}] = \mathbf{I}$. We can always simulate the SDE to sample from $p_{0t}(\mathbf{x}(t) \mid \mathbf{x}(0))$, and solve Eq. (19) to train the time-dependent score-based model $\mathbf{s}_{\boldsymbol{\theta}}(\mathbf{x}, t)$.

## B  VE, VP AND SUB-VP SDEs

Below we provide detailed derivations to show that the noise perturbations of SMLD and DDPM are discretizations of the Variance Exploding (VE) and Variance Preserving (VP) SDEs respectively. We additionally introduce sub-VP SDEs, a modification to VP SDEs that often achieves better performance in both sample quality and likelihoods.

First, when using a total of $N$ noise scales, each perturbation kernel $p_{\sigma_i}(\mathbf{x} \mid \mathbf{x}_0)$ of SMLD can be derived from the following Markov chain:

$$\mathbf{x}_i = \mathbf{x}_{i-1} + \sqrt{\sigma_i^2 - \sigma_{i-1}^2}\mathbf{z}_{i-1}, \quad i = 1, \cdots, N, \tag{20}$$

where $\mathbf{z}_{i-1} \sim \mathcal{N}(\mathbf{0}, \mathbf{I})$, $\mathbf{x}_0 \sim p_{\text{data}}$, and we have introduced $\sigma_0 = 0$ to simplify the notation. In the limit of $N \to \infty$, the Markov chain $\{\mathbf{x}_i\}_{i=1}^N$ becomes a continuous stochastic process $\{\mathbf{x}(t)\}_{t=0}^1$, $\{\sigma_i\}_{i=1}^N$ becomes a function $\sigma(t)$, and $\mathbf{z}_i$ becomes $\mathbf{z}(t)$, where we have used a continuous time variable $t \in [0, 1]$ for indexing, rather than an integer $i \in \{1, 2, \cdots, N\}$. Let $\mathbf{x}\left(\frac{i}{N}\right) = \mathbf{x}_i$, $\sigma\left(\frac{i}{N}\right) = \sigma_i$, and $\mathbf{z}\left(\frac{i}{N}\right) = \mathbf{z}_i$ for $i = 1, 2, \cdots, N$. We can rewrite Eq. (20) as follows with $\Delta t = \frac{1}{N}$ and $t \in \{0, \frac{1}{N}, \cdots, \frac{N-1}{N}\}$:

$$\mathbf{x}(t + \Delta t) = \mathbf{x}(t) + \sqrt{\sigma^2(t + \Delta t) - \sigma^2(t)}\,\mathbf{z}(t) \approx \mathbf{x}(t) + \sqrt{\frac{\mathrm{d}\left[\sigma^2(t)\right]}{\mathrm{d}t}\Delta t}\,\mathbf{z}(t),$$

where the approximate equality holds when $\Delta t \ll 1$. In the limit of $\Delta t \to 0$, this converges to

$$\mathrm{d}\mathbf{x} = \sqrt{\frac{\mathrm{d}\left[\sigma^2(t)\right]}{\mathrm{d}t}}\mathrm{d}\mathbf{w}, \tag{21}$$

which is the VE SDE.

For the perturbation kernels $\{p_{\alpha_i}(\mathbf{x} \mid \mathbf{x}_0)\}_{i=1}^N$ used in DDPM, the discrete Markov chain is

$$\mathbf{x}_i = \sqrt{1 - \beta_i}\mathbf{x}_{i-1} + \sqrt{\beta_i}\mathbf{z}_{i-1}, \quad i = 1, \cdots, N, \tag{22}$$

where $\mathbf{z}_{i-1} \sim \mathcal{N}(\mathbf{0}, \mathbf{I})$. To obtain the limit of this Markov chain when $N \to \infty$, we define an auxiliary set of noise scales $\{\bar{\beta}_i = N\beta_i\}_{i=1}^N$, and re-write Eq. (22) as below

$$\mathbf{x}_i = \sqrt{1 - \frac{\bar{\beta}_i}{N}}\mathbf{x}_{i-1} + \sqrt{\frac{\bar{\beta}_i}{N}}\mathbf{z}_{i-1}, \quad i = 1, \cdots, N. \tag{23}$$

In the limit of $N \to \infty$, $\{\bar{\beta}_i\}_{i=1}^N$ becomes a function $\beta(t)$ indexed by $t \in [0, 1]$. Let $\beta\left(\frac{i}{N}\right) = \bar{\beta}_i$, $\mathbf{x}(\frac{i}{N}) = \mathbf{x}_i$, $\mathbf{z}(\frac{i}{N}) = \mathbf{z}_i$. We can rewrite the Markov chain Eq. (23) as the following with $\Delta t = \frac{1}{N}$ and $t \in \{0, 1, \cdots, \frac{N-1}{N}\}$:

$$\begin{aligned}
\mathbf{x}(t + \Delta t) &= \sqrt{1 - \beta(t + \Delta t)\Delta t}\,\mathbf{x}(t) + \sqrt{\beta(t + \Delta t)\Delta t}\,\mathbf{z}(t) \\
&\approx \mathbf{x}(t) - \frac{1}{2}\beta(t + \Delta t)\Delta t\,\mathbf{x}(t) + \sqrt{\beta(t + \Delta t)\Delta t}\,\mathbf{z}(t) \\
&\approx \mathbf{x}(t) - \frac{1}{2}\beta(t)\Delta t\,\mathbf{x}(t) + \sqrt{\beta(t)\Delta t}\,\mathbf{z}(t),
\end{aligned} \tag{24}$$

where the approximate equality holds when $\Delta t \ll 1$. Therefore, in the limit of $\Delta t \to 0$, Eq. (24) converges to the following VP SDE:

$$\mathrm{d}\mathbf{x} = -\frac{1}{2}\beta(t)\mathbf{x}\,\mathrm{d}t + \sqrt{\beta(t)}\,\mathrm{d}\mathbf{w}. \tag{25}$$

So far, we have demonstrated that the noise perturbations used in SMLD and DDPM correspond to discretizations of VE and VP SDEs respectively. The VE SDE always yields a process with exploding variance when $t \to \infty$. In contrast, the VP SDE yields a process with bounded variance. In addition, the process has a constant unit variance for all $t \in [0, \infty)$ when $p(\mathbf{x}(0))$ has a unit variance. Since the VP SDE has affine drift and diffusion coefficients, we can use Eq. (5.51) in Särkkä & Solin (2019) to obtain an ODE that governs the evolution of variance

$$\frac{\mathrm{d}\mathbf{\Sigma}_{\text{VP}}(t)}{\mathrm{d}t} = \beta(t)(\mathbf{I} - \mathbf{\Sigma}_{\text{VP}}(t)),$$

where $\boldsymbol{\Sigma}_{\mathrm{VP}}(t) := \mathrm{Cov}[\mathbf{x}(t)]$ for $\{\mathbf{x}(t)\}_{t=0}^1$ obeying a VP SDE. Solving this ODE, we obtain

$$\boldsymbol{\Sigma}_{\mathrm{VP}}(t) = \mathbf{I} + e^{\int_0^t -\beta(s)\mathrm{d}s}(\boldsymbol{\Sigma}_{\mathrm{VP}}(0) - \mathbf{I}), \tag{26}$$

from which it is clear that the variance $\boldsymbol{\Sigma}_{\mathrm{VP}}(t)$ is always bounded given $\boldsymbol{\Sigma}_{\mathrm{VP}}(0)$. Moreover, $\boldsymbol{\Sigma}_{\mathrm{VP}}(t) \equiv \mathbf{I}$ if $\boldsymbol{\Sigma}_{\mathrm{VP}}(0) = \mathbf{I}$. Due to this difference, we name Eq. (9) as the *Variance Exploding (VE) SDE*, and Eq. (11) the *Variance Preserving (VP) SDE*.

Inspired by the VP SDE, we propose a new SDE called the *sub-VP SDE*, namely

$$\mathrm{d}\mathbf{x} = -\frac{1}{2}\beta(t)\mathbf{x}\,\mathrm{d}t + \sqrt{\beta(t)(1 - e^{-2\int_0^t \beta(s)\mathrm{d}s})}\mathrm{d}\mathbf{w}. \tag{27}$$

Following standard derivations, it is straightforward to show that $\mathbb{E}[\mathbf{x}(t)]$ is the same for both VP and sub-VP SDEs; the variance function of sub-VP SDEs is different, given by

$$\boldsymbol{\Sigma}_{\mathrm{sub\text{-}VP}}(t) = \mathbf{I} + e^{-2\int_0^t \beta(s)\mathrm{d}s}\mathbf{I} + e^{-\int_0^t \beta(s)\mathrm{d}s}(\boldsymbol{\Sigma}_{\mathrm{sub\text{-}VP}}(0) - 2\mathbf{I}), \tag{28}$$

where $\boldsymbol{\Sigma}_{\mathrm{sub\text{-}VP}}(t) := \mathrm{Cov}[\mathbf{x}(t)]$ for a process $\{\mathbf{x}(t)\}_{t=0}^1$ obtained by solving Eq. (27). In addition, we observe that (i) $\boldsymbol{\Sigma}_{\mathrm{sub\text{-}VP}}(t) \preccurlyeq \boldsymbol{\Sigma}_{\mathrm{VP}}(t)$ for all $t \geqslant 0$ with $\boldsymbol{\Sigma}_{\mathrm{sub\text{-}VP}}(0) = \boldsymbol{\Sigma}_{\mathrm{VP}}(0)$ and shared $\beta(s)$; and (ii) $\lim_{t\to\infty} \boldsymbol{\Sigma}_{\mathrm{sub\text{-}VP}}(t) = \lim_{t\to\infty} \boldsymbol{\Sigma}_{\mathrm{VP}}(t) = \mathbf{I}$ if $\lim_{t\to\infty} \int_0^t \beta(s)\mathrm{d}s = \infty$. The former is why we name Eq. (27) the sub-VP SDE—its variance is always upper bounded by the corresponding VP SDE. The latter justifies the use of sub-VP SDEs for score-based generative modeling, since they can perturb any data distribution to standard Gaussian under suitable conditions, just like VP SDEs.

VE, VP and sub-VP SDEs all have affine drift coefficients. Therefore, their perturbation kernels $p_{0t}(\mathbf{x}(t) \mid \mathbf{x}(0))$ are all Gaussian and can be computed with Eqs. (5.50) and (5.51) in Särkkä & Solin (2019):

$$p_{0t}(\mathbf{x}(t) \mid \mathbf{x}(0)) = \begin{cases} \mathcal{N}\big(\mathbf{x}(t); \mathbf{x}(0), [\sigma^2(t) - \sigma^2(0)]\mathbf{I}\big), & \text{(VE SDE)} \\ \mathcal{N}\big(\mathbf{x}(t); \mathbf{x}(0)e^{-\frac{1}{2}\int_0^t \beta(s)\mathrm{d}s}, \mathbf{I} - \mathbf{I}e^{-\int_0^t \beta(s)\mathrm{d}s}\big) & \text{(VP SDE)} \\ \mathcal{N}\big(\mathbf{x}(t); \mathbf{x}(0)e^{-\frac{1}{2}\int_0^t \beta(s)\mathrm{d}s}, [1 - e^{-\int_0^t \beta(s)\mathrm{d}s}]^2\mathbf{I}\big) & \text{(sub-VP SDE)} \end{cases} . \tag{29}$$

As a result, all SDEs introduced here can be efficiently trained with the objective in Eq. (7).

## C  SDEs IN THE WILD

Below we discuss concrete instantiations of VE and VP SDEs whose discretizations yield SMLD and DDPM models, and the specific sub-VP SDE used in our experiments. In SMLD, the noise scales $\{\sigma_i\}_{i=1}^N$ is typically a geometric sequence where $\sigma_{\min}$ is fixed to $0.01$ and $\sigma_{\max}$ is chosen according to Technique 1 in Song & Ermon (2020). Usually, SMLD models normalize image inputs to the range $[0, 1]$. Since $\{\sigma_i\}_{i=1}^N$ is a geometric sequence, we have $\sigma(\frac{i}{N}) = \sigma_i = \sigma_{\min}\left(\frac{\sigma_{\max}}{\sigma_{\min}}\right)^{\frac{i-1}{N-1}}$ for $i = 1, 2, \cdots, N$. In the limit of $N \to \infty$, we have $\sigma(t) = \sigma_{\min}\left(\frac{\sigma_{\max}}{\sigma_{\min}}\right)^t$ for $t \in (0, 1]$. The corresponding VE SDE is

$$\mathrm{d}\mathbf{x} = \sigma_{\min}\left(\frac{\sigma_{\max}}{\sigma_{\min}}\right)^t \sqrt{2\log\frac{\sigma_{\max}}{\sigma_{\min}}}\mathrm{d}\mathbf{w}, \quad t \in (0, 1], \tag{30}$$

and the perturbation kernel can be derived via Eq. (29):

$$p_{0t}(\mathbf{x}(t) \mid \mathbf{x}(0)) = \mathcal{N}\left(\mathbf{x}(t); \mathbf{x}(0), \sigma_{\min}^2\left(\frac{\sigma_{\max}}{\sigma_{\min}}\right)^{2t}\mathbf{I}\right), \quad t \in (0, 1]. \tag{31}$$

There is one subtlety when $t = 0$: by definition, $\sigma(0) = \sigma_0 = 0$ (following the convention in Eq. (20)), but $\sigma(0^+) := \lim_{t\to 0^+} \sigma(t) = \sigma_{\min} \neq 0$. In other words, $\sigma(t)$ for SMLD is not differentiable since $\sigma(0) \neq \sigma(0^+)$, causing the VE SDE in Eq. (21) undefined for $t = 0$. In practice, we bypass this issue by always solving the SDE and its associated probability flow ODE in the range $t \in [\epsilon, 1]$ for some small constant $\epsilon > 0$, and we use $\epsilon = 10^{-5}$ in our VE SDE experiments.

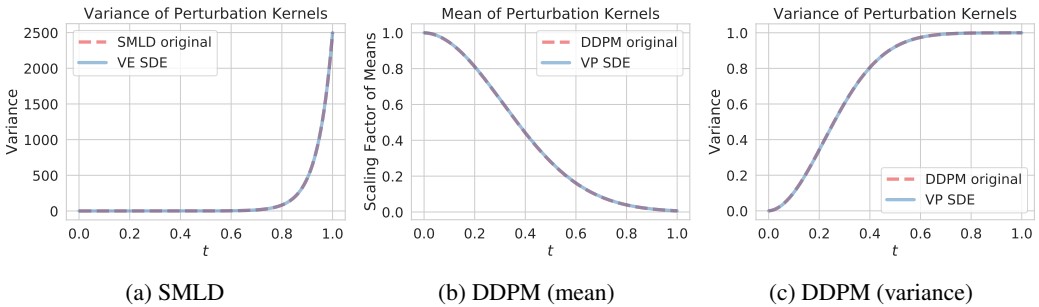

(a) SMLD            (b) DDPM (mean)            (c) DDPM (variance)

Figure 5: Discrete-time perturbation kernels and our continuous generalizations match each other almost exactly. (a) compares the variance of perturbation kernels for SMLD and VE SDE; (b) compares the scaling factors of means of perturbation kernels for DDPM and VP SDE; and (c) compares the variance of perturbation kernels for DDPM and VP SDE.

For DDPM models, $\{\beta_i\}_{i=1}^N$ is typically an arithmetic sequence where $\beta_i = \frac{\bar{\beta}_{\min}}{N} + \frac{i-1}{N(N-1)}(\bar{\beta}_{\max} - \bar{\beta}_{\min})$ for $i = 1, 2, \cdots, N$. Therefore, $\beta(t) = \bar{\beta}_{\min} + t(\bar{\beta}_{\max} - \bar{\beta}_{\min})$ for $t \in [0, 1]$ in the limit of $N \to \infty$. This corresponds to the following instantiation of the VP SDE:

$$\mathrm{d}\mathbf{x} = -\frac{1}{2}(\bar{\beta}_{\min} + t(\bar{\beta}_{\max} - \bar{\beta}_{\min}))\mathbf{x}\mathrm{d}t + \sqrt{\bar{\beta}_{\min} + t(\bar{\beta}_{\max} - \bar{\beta}_{\min})}\mathrm{d}\mathbf{w}, \quad t \in [0, 1], \quad (32)$$

where $\mathbf{x}(0) \sim p_{\text{data}}(\mathbf{x})$. In our experiments, we let $\bar{\beta}_{\min} = 0.1$ and $\bar{\beta}_{\max} = 20$ to match the settings in Ho et al. (2020). The perturbation kernel is given by

$$p_{0t}(\mathbf{x}(t) \mid \mathbf{x}(0))$$
$$= \mathcal{N}\left(\mathbf{x}(t); e^{-\frac{1}{4}t^2(\bar{\beta}_{\max} - \bar{\beta}_{\min}) - \frac{1}{2}t\bar{\beta}_{\min}}\mathbf{x}(0), \mathbf{I} - \mathbf{I}e^{-\frac{1}{2}t^2(\bar{\beta}_{\max} - \bar{\beta}_{\min}) - t\bar{\beta}_{\min}}\right), \quad t \in [0, 1]. \quad (33)$$

For DDPM, there is no discontinuity issue with the corresponding VP SDE; yet, there are numerical instability issues for training and sampling at $t = 0$, due to the vanishing variance of $\mathbf{x}(t)$ as $t \to 0$. Therefore, same as the VE SDE, we restrict computation to $t \in [\epsilon, 1]$ for a small $\epsilon > 0$. For sampling, we choose $\epsilon = 10^{-3}$ so that the variance of $\mathbf{x}(\epsilon)$ in VP SDE matches the variance of $\mathbf{x}_1$ in DDPM; for training and likelihood computation, we adopt $\epsilon = 10^{-5}$ which empirically gives better results.

As a sanity check for our SDE generalizations to SMLD and DDPM, we compare the perturbation kernels of SDEs and original discrete Markov chains in Fig. 5. The SMLD and DDPM models both use $N = 1000$ noise scales. For SMLD, we only need to compare the variances of perturbation kernels since means are the same by definition. For DDPM, we compare the scaling factors of means and the variances. As demonstrated in Fig. 5, the discrete perturbation kernels of original SMLD and DDPM models align well with perturbation kernels derived from VE and VP SDEs.

For sub-VP SDEs, we use exactly the same $\beta(t)$ as VP SDEs. This leads to the following perturbation kernel

$$p_{0t}(\mathbf{x}(t) \mid \mathbf{x}(0))$$
$$= \mathcal{N}\left(\mathbf{x}(t); e^{-\frac{1}{4}t^2(\bar{\beta}_{\max} - \bar{\beta}_{\min}) - \frac{1}{2}t\bar{\beta}_{\min}}\mathbf{x}(0), [1 - e^{-\frac{1}{2}t^2(\bar{\beta}_{\max} - \bar{\beta}_{\min}) - t\bar{\beta}_{\min}}]^2\mathbf{I}\right), \quad t \in [0, 1]. \quad (34)$$

We also restrict numerical computation to the same interval of $[\epsilon, 1]$ as VP SDEs.

Empirically, we observe that smaller $\epsilon$ generally yields better likelihood values for all SDEs. For sampling, it is important to use an appropriate $\epsilon$ for better Inception scores and FIDs, although samples across different $\epsilon$ look visually the same to human eyes.

## D    PROBABILITY FLOW ODE

### D.1    DERIVATION

The idea of probability flow ODE is inspired by Maoutsa et al. (2020), and one can find the derivation of a simplified case therein. Below we provide a derivation for the fully general ODE in Eq. (17). We

consider the SDE in Eq. (15), which possesses the following form:

$$d\mathbf{x} = \mathbf{f}(\mathbf{x}, t)dt + \mathbf{G}(\mathbf{x}, t)d\mathbf{w},$$

where $\mathbf{f}(\cdot, t) : \mathbb{R}^d \to \mathbb{R}^d$ and $\mathbf{G}(\cdot, t) : \mathbb{R}^d \to \mathbb{R}^{d \times d}$. The marginal probability density $p_t(\mathbf{x}(t))$ evolves according to Kolmogorov's forward equation (Fokker-Planck equation) (Øksendal, 2003)

$$\frac{\partial p_t(\mathbf{x})}{\partial t} = -\sum_{i=1}^{d} \frac{\partial}{\partial x_i}[f_i(\mathbf{x}, t)p_t(\mathbf{x})] + \frac{1}{2}\sum_{i=1}^{d}\sum_{j=1}^{d} \frac{\partial^2}{\partial x_i \partial x_j}\Big[\sum_{k=1}^{d} G_{ik}(\mathbf{x}, t)G_{jk}(\mathbf{x}, t)p_t(\mathbf{x})\Big]. \quad (35)$$

We can easily rewrite Eq. (35) to obtain

$$\frac{\partial p_t(\mathbf{x})}{\partial t} = -\sum_{i=1}^{d} \frac{\partial}{\partial x_i}[f_i(\mathbf{x}, t)p_t(\mathbf{x})] + \frac{1}{2}\sum_{i=1}^{d}\sum_{j=1}^{d} \frac{\partial^2}{\partial x_i \partial x_j}\Big[\sum_{k=1}^{d} G_{ik}(\mathbf{x}, t)G_{jk}(\mathbf{x}, t)p_t(\mathbf{x})\Big]$$

$$= -\sum_{i=1}^{d} \frac{\partial}{\partial x_i}[f_i(\mathbf{x}, t)p_t(\mathbf{x})] + \frac{1}{2}\sum_{i=1}^{d} \frac{\partial}{\partial x_i}\Big[\sum_{j=1}^{d} \frac{\partial}{\partial x_j}\Big[\sum_{k=1}^{d} G_{ik}(\mathbf{x}, t)G_{jk}(\mathbf{x}, t)p_t(\mathbf{x})\Big]\Big]. \quad (36)$$

Note that

$$\sum_{j=1}^{d} \frac{\partial}{\partial x_j}\Big[\sum_{k=1}^{d} G_{ik}(\mathbf{x}, t)G_{jk}(\mathbf{x}, t)p_t(\mathbf{x})\Big]$$

$$= \sum_{j=1}^{d} \frac{\partial}{\partial x_j}\Big[\sum_{k=1}^{d} G_{ik}(\mathbf{x}, t)G_{jk}(\mathbf{x}, t)\Big]p_t(\mathbf{x}) + \sum_{j=1}^{d}\sum_{k=1}^{d} G_{ik}(\mathbf{x}, t)G_{jk}(\mathbf{x}, t)p_t(\mathbf{x})\frac{\partial}{\partial x_j}\log p_t(\mathbf{x})$$

$$= p_t(\mathbf{x})\nabla \cdot [\mathbf{G}(\mathbf{x}, t)\mathbf{G}(\mathbf{x}, t)^\mathsf{T}] + p_t(\mathbf{x})\mathbf{G}(\mathbf{x}, t)\mathbf{G}(\mathbf{x}, t)^\mathsf{T}\nabla_\mathbf{x}\log p_t(\mathbf{x}),$$

based on which we can continue the rewriting of Eq. (36) to obtain

$$\frac{\partial p_t(\mathbf{x})}{\partial t} = -\sum_{i=1}^{d} \frac{\partial}{\partial x_i}[f_i(\mathbf{x}, t)p_t(\mathbf{x})] + \frac{1}{2}\sum_{i=1}^{d} \frac{\partial}{\partial x_i}\Big[\sum_{j=1}^{d} \frac{\partial}{\partial x_j}\Big[\sum_{k=1}^{d} G_{ik}(\mathbf{x}, t)G_{jk}(\mathbf{x}, t)p_t(\mathbf{x})\Big]\Big]$$

$$= -\sum_{i=1}^{d} \frac{\partial}{\partial x_i}[f_i(\mathbf{x}, t)p_t(\mathbf{x})]$$

$$+ \frac{1}{2}\sum_{i=1}^{d} \frac{\partial}{\partial x_i}\Big[p_t(\mathbf{x})\nabla \cdot [\mathbf{G}(\mathbf{x}, t)\mathbf{G}(\mathbf{x}, t)^\mathsf{T}] + p_t(\mathbf{x})\mathbf{G}(\mathbf{x}, t)\mathbf{G}(\mathbf{x}, t)^\mathsf{T}\nabla_\mathbf{x}\log p_t(\mathbf{x})\Big]$$

$$= -\sum_{i=1}^{d} \frac{\partial}{\partial x_i}\Big\{f_i(\mathbf{x}, t)p_t(\mathbf{x})$$

$$- \frac{1}{2}\Big[\nabla \cdot [\mathbf{G}(\mathbf{x}, t)\mathbf{G}(\mathbf{x}, t)^\mathsf{T}] + \mathbf{G}(\mathbf{x}, t)\mathbf{G}(\mathbf{x}, t)^\mathsf{T}\nabla_\mathbf{x}\log p_t(\mathbf{x})\Big]p_t(\mathbf{x})\Big\}$$

$$= -\sum_{i=1}^{d} \frac{\partial}{\partial x_i}[\tilde{f}_i(\mathbf{x}, t)p_t(\mathbf{x})], \quad (37)$$

where we define

$$\tilde{\mathbf{f}}(\mathbf{x}, t) := \mathbf{f}(\mathbf{x}, t) - \frac{1}{2}\nabla \cdot [\mathbf{G}(\mathbf{x}, t)\mathbf{G}(\mathbf{x}, t)^\mathsf{T}] - \frac{1}{2}\mathbf{G}(\mathbf{x}, t)\mathbf{G}(\mathbf{x}, t)^\mathsf{T}\nabla_\mathbf{x}\log p_t(\mathbf{x}).$$

Inspecting Eq. (37), we observe that it equals Kolmogorov's forward equation of the following SDE with $\tilde{\mathbf{G}}(\mathbf{x}, t) := \mathbf{0}$ (Kolmogorov's forward equation in this case is also known as the Liouville equation.)

$$d\mathbf{x} = \tilde{\mathbf{f}}(\mathbf{x}, t)dt + \tilde{\mathbf{G}}(\mathbf{x}, t)d\mathbf{w},$$

which is essentially an ODE:

$$d\mathbf{x} = \tilde{\mathbf{f}}(\mathbf{x}, t)dt,$$

same as the probability flow ODE given by Eq. (17). Therefore, we have shown that the probability flow ODE Eq. (17) induces the same marginal probability density $p_t(\mathbf{x})$ as the SDE in Eq. (15).

## D.2 LIKELIHOOD COMPUTATION

The probability flow ODE in Eq. (17) has the following form when we replace the score $\nabla_{\mathbf{x}} \log p_t(\mathbf{x})$ with the time-dependent score-based model $\mathbf{s}_{\boldsymbol{\theta}}(\mathbf{x}, t)$:

$$\mathrm{d}\mathbf{x} = \underbrace{\left\{ \mathbf{f}(\mathbf{x}, t) - \frac{1}{2} \nabla \cdot [\mathbf{G}(\mathbf{x}, t)\mathbf{G}(\mathbf{x}, t)^{\mathsf{T}}] - \frac{1}{2}\mathbf{G}(\mathbf{x}, t)\mathbf{G}(\mathbf{x}, t)^{\mathsf{T}}\mathbf{s}_{\boldsymbol{\theta}}(\mathbf{x}, t) \right\}}_{=:\tilde{\mathbf{f}}_{\boldsymbol{\theta}}(\mathbf{x}, t)} \mathrm{d}t. \tag{38}$$

With the instantaneous change of variables formula (Chen et al., 2018), we can compute the log-likelihood of $p_0(\mathbf{x})$ using

$$\log p_0(\mathbf{x}(0)) = \log p_T(\mathbf{x}(T)) + \int_0^T \nabla \cdot \tilde{\mathbf{f}}_{\boldsymbol{\theta}}(\mathbf{x}(t), t)\mathrm{d}t, \tag{39}$$

where the random variable $\mathbf{x}(t)$ as a function of $t$ can be obtained by solving the probability flow ODE in Eq. (38). In many cases computing $\nabla \cdot \tilde{\mathbf{f}}_{\boldsymbol{\theta}}(\mathbf{x}, t)$ is expensive, so we follow Grathwohl et al. (2018) to estimate it with the Skilling-Hutchinson trace estimator (Skilling, 1989; Hutchinson, 1990). In particular, we have

$$\nabla \cdot \tilde{\mathbf{f}}_{\boldsymbol{\theta}}(\mathbf{x}, t) = \mathbb{E}_{p(\boldsymbol{\epsilon})}[\boldsymbol{\epsilon}^{\mathsf{T}}\nabla\tilde{\mathbf{f}}_{\boldsymbol{\theta}}(\mathbf{x}, t)\boldsymbol{\epsilon}], \tag{40}$$

where $\nabla\tilde{\mathbf{f}}_{\boldsymbol{\theta}}$ denotes the Jacobian of $\tilde{\mathbf{f}}_{\boldsymbol{\theta}}(\cdot, t)$, and the random variable $\boldsymbol{\epsilon}$ satisfies $\mathbb{E}_{p(\boldsymbol{\epsilon})}[\boldsymbol{\epsilon}] = \mathbf{0}$ and $\mathrm{Cov}_{p(\boldsymbol{\epsilon})}[\boldsymbol{\epsilon}] = \mathbf{I}$. The vector-Jacobian product $\boldsymbol{\epsilon}^{\mathsf{T}}\nabla\tilde{\mathbf{f}}_{\boldsymbol{\theta}}(\mathbf{x}, t)$ can be efficiently computed using reverse-mode automatic differentiation, at approximately the same cost as evaluating $\tilde{\mathbf{f}}_{\boldsymbol{\theta}}(\mathbf{x}, t)$. As a result, we can sample $\boldsymbol{\epsilon} \sim p(\boldsymbol{\epsilon})$ and then compute an efficient unbiased estimate to $\nabla \cdot \tilde{\mathbf{f}}_{\boldsymbol{\theta}}(\mathbf{x}, t)$ using $\boldsymbol{\epsilon}^{\mathsf{T}}\nabla\tilde{\mathbf{f}}_{\boldsymbol{\theta}}(\mathbf{x}, t)\boldsymbol{\epsilon}$. Since this estimator is unbiased, we can attain an arbitrarily small error by averaging over a sufficient number of runs. Therefore, by applying the Skilling-Hutchinson estimator Eq. (40) to Eq. (39), we can compute the log-likelihood to any accuracy.

In our experiments, we use the RK45 ODE solver (Dormand & Prince, 1980) provided by `scipy.integrate.solve_ivp` in all cases. The bits/dim values in Table 2 are computed with `atol=1e-5` and `rtol=1e-5`, same as Grathwohl et al. (2018). To give the likelihood results of our models in Table 2, we average the bits/dim obtained on the test dataset over five different runs with $\epsilon = 10^{-5}$ (see definition of $\epsilon$ in Appendix C).

## D.3 PROBABILITY FLOW SAMPLING

Suppose we have a forward SDE

$$\mathrm{d}\mathbf{x} = \mathbf{f}(\mathbf{x}, t)\mathrm{d}t + \mathbf{G}(t)\mathrm{d}\mathbf{w},$$

and one of its discretization

$$\mathbf{x}_{i+1} = \mathbf{x}_i + \mathbf{f}_i(\mathbf{x}_i) + \mathbf{G}_i\mathbf{z}_i, \quad i = 0, 1, \cdots, N-1, \tag{41}$$

where $\mathbf{z}_i \sim \mathcal{N}(\mathbf{0}, \mathbf{I})$. We assume the discretization schedule of time is fixed beforehand, and thus we absorb the dependency on $\Delta t$ into the notations of $\mathbf{f}_i$ and $\mathbf{G}_i$. Using Eq. (17), we can obtain the following probability flow ODE:

$$\mathrm{d}\mathbf{x} = \left\{ \mathbf{f}(\mathbf{x}, t) - \frac{1}{2}\mathbf{G}(t)\mathbf{G}(t)^{\mathsf{T}}\nabla_{\mathbf{x}} \log p_t(\mathbf{x}) \right\} \mathrm{d}t. \tag{42}$$

We may employ any numerical method to integrate the probability flow ODE backwards in time for sample generation. In particular, we propose a discretization in a similar functional form to Eq. (41):

$$\mathbf{x}_i = \mathbf{x}_{i+1} - \mathbf{f}_{i+1}(\mathbf{x}_{i+1}) + \frac{1}{2}\mathbf{G}_{i+1}\mathbf{G}_{i+1}^{\mathsf{T}}\mathbf{s}_{\boldsymbol{\theta}*}(\mathbf{x}_{i+1}, i+1), \quad i = 0, 1, \cdots, N-1,$$

where the score-based model $\mathbf{s}_{\boldsymbol{\theta}*}(\mathbf{x}_i, i)$ is conditioned on the iteration number $i$. This is a deterministic iteration rule. Unlike reverse diffusion samplers or ancestral sampling, there is no additional randomness once the initial sample $\mathbf{x}_N$ is obtained from the prior distribution. When applied to SMLD models, we can get the following iteration rule for probability flow sampling:

$$\mathbf{x}_i = \mathbf{x}_{i+1} + \frac{1}{2}(\sigma_{i+1}^2 - \sigma_i^2)\mathbf{s}_{\boldsymbol{\theta}*}(\mathbf{x}_{i+1}, \sigma_{i+1}), \quad i = 0, 1, \cdots, N-1. \tag{43}$$

Similarly, for DDPM models, we have

$$\mathbf{x}_i = (2 - \sqrt{1 - \beta_{i+1}})\mathbf{x}_{i+1} + \frac{1}{2}\beta_{i+1}\mathbf{s}_{\boldsymbol{\theta}*}(\mathbf{x}_{i+1}, i+1), \quad i = 0, 1, \cdots, N-1. \tag{44}$$

### D.4 SAMPLING WITH BLACK-BOX ODE SOLVERS

For producing figures in Fig. 3, we use a DDPM model trained on $256 \times 256$ CelebA-HQ with the same settings in Ho et al. (2020). All FID scores of our models in Table 2 are computed on samples from the RK45 ODE solver implemented in `scipy.integrate.solve_ivp` with `atol=1e-5` and `rtol=1e-5`. We use $\epsilon = 10^{-5}$ for VE SDEs and $\epsilon = 10^{-3}$ for VP SDEs (see also Appendix C).

Aside from the interpolation results in Fig. 3, we demonstrate more examples of latent space manipulation in Fig. 6, including interpolation and temperature scaling. The model tested here is a DDPM model trained with the same settings in Ho et al. (2020).

Although solvers for the probability flow ODE allow fast sampling, their samples typically have higher (worse) FID scores than those from SDE solvers if no corrector is used. We have this empirical observation for both the discretization strategy in Appendix D.3, and black-box ODE solvers introduced above. Moreover, the performance of probability flow ODE samplers depends on the choice of the SDE—their sample quality for VE SDEs is much worse than VP SDEs especially for high-dimensional data.

### D.5 UNIQUELY IDENTIFIABLE ENCODING

As a sanity check, we train two models (denoted as "Model A" and "Model B") with different architectures using the VE SDE on CIFAR-10. Here Model A is an NCSN++ model with 4 layers per resolution trained using the continuous objective in Eq. (7), and Model B is all the same except that it uses 8 layers per resolution. Model definitions are in Appendix H.

We report the latent codes obtained by Model A and Model B for a random CIFAR-10 image in Fig. 7. In Fig. 8, we show the dimension-wise differences and correlation coefficients between latent encodings on a total of 16 CIFAR-10 images. Our results demonstrate that for the same inputs, Model A and Model B provide encodings that are close in every dimension, despite having different model architectures and training runs.

## E REVERSE DIFFUSION SAMPLING

Given a forward SDE

$$\mathrm{d}\mathbf{x} = \mathbf{f}(\mathbf{x}, t)\mathrm{d}t + \mathbf{G}(t)\mathrm{d}\mathbf{w},$$

and suppose the following iteration rule is a discretization of it:

$$\mathbf{x}_{i+1} = \mathbf{x}_i + \mathbf{f}_i(\mathbf{x}_i) + \mathbf{G}_i\mathbf{z}_i, \quad i = 0, 1, \cdots, N-1 \tag{45}$$

where $\mathbf{z}_i \sim \mathcal{N}(\mathbf{0}, \mathbf{I})$. Here we assume the discretization schedule of time is fixed beforehand, and thus we can absorb it into the notations of $\mathbf{f}_i$ and $\mathbf{G}_i$.

Based on Eq. (45), we propose to discretize the reverse-time SDE

$$\mathrm{d}\mathbf{x} = [\mathbf{f}(\mathbf{x}, t) - \mathbf{G}(t)\mathbf{G}(t)^\mathsf{T}\nabla_\mathbf{x} \log p_t(\mathbf{x})]\mathrm{d}t + \mathbf{G}(t)\mathrm{d}\bar{\mathbf{w}},$$

with a similar functional form, which gives the following iteration rule for $i \in \{0, 1, \cdots, N-1\}$:

$$\mathbf{x}_i = \mathbf{x}_{i+1} - \mathbf{f}_{i+1}(\mathbf{x}_{i+1}) + \mathbf{G}_{i+1}\mathbf{G}_{i+1}^\mathsf{T}\mathbf{s}_{\boldsymbol{\theta}*}(\mathbf{x}_{i+1}, i+1) + \mathbf{G}_{i+1}\mathbf{z}_{i+1}, \tag{46}$$

where our trained score-based model $\mathbf{s}_{\boldsymbol{\theta}*}(\mathbf{x}_i, i)$ is conditioned on iteration number $i$.

When applying Eq. (46) to Eqs. (10) and (20), we obtain a new set of numerical solvers for the reverse-time VE and VP SDEs, resulting in sampling algorithms as shown in the "predictor" part of Algorithms 2 and 3. We name these sampling methods (that are based on the discretization strategy in Eq. (46)) *reverse diffusion samplers*.

As expected, the ancestral sampling of DDPM (Ho et al., 2020) (Eq. (4)) matches its reverse diffusion counterpart when $\beta_i \to 0$ for all $i$ (which happens when $\Delta t \to 0$ since $\beta_i = \bar{\beta}_i\Delta t$, see Appendix B),

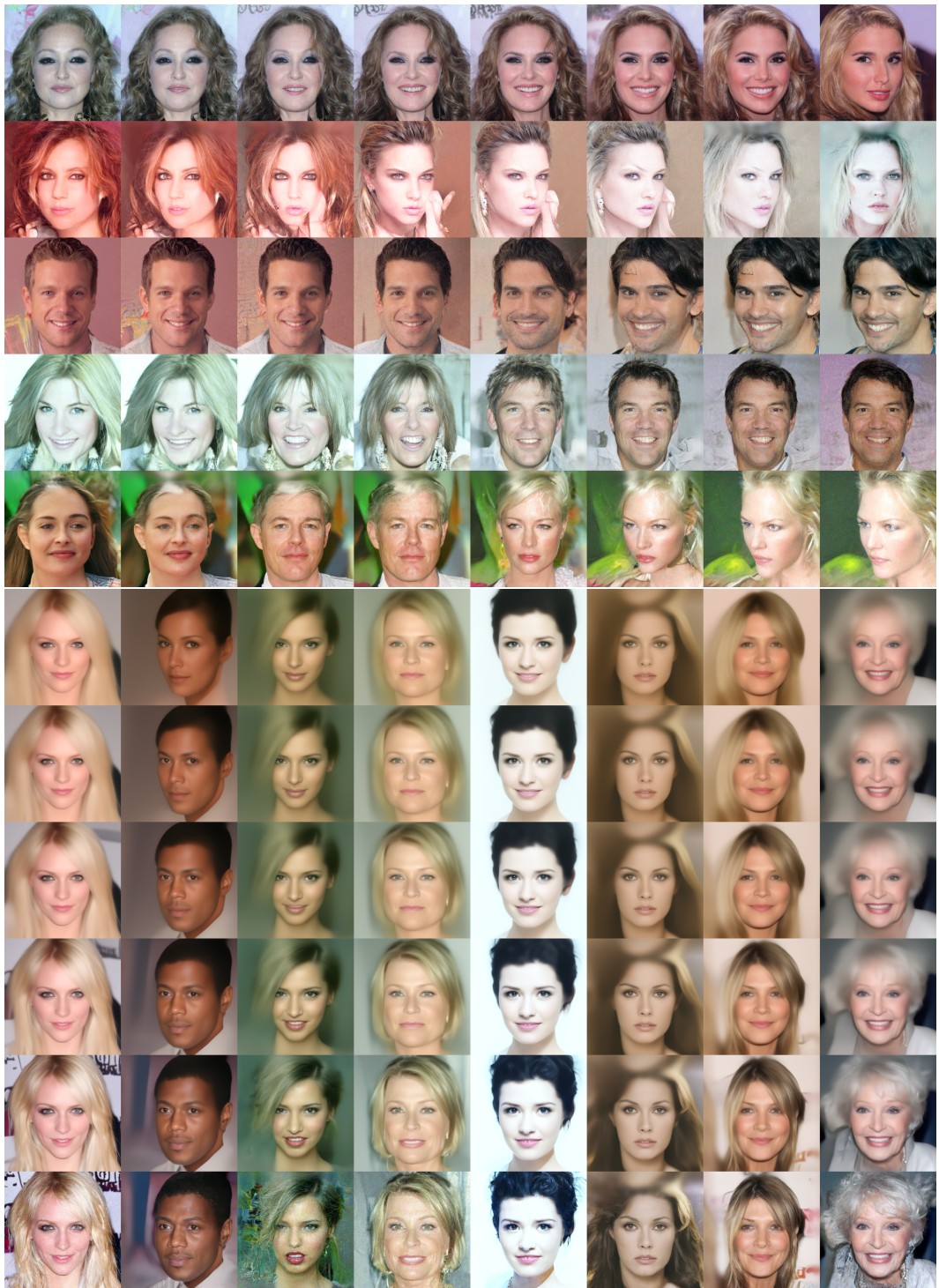

Figure 6: Samples from the probability flow ODE for VP SDE on $256 \times 256$ CelebA-HQ. Top: spherical interpolations between random samples. Bottom: temperature rescaling (reducing norm of embedding).

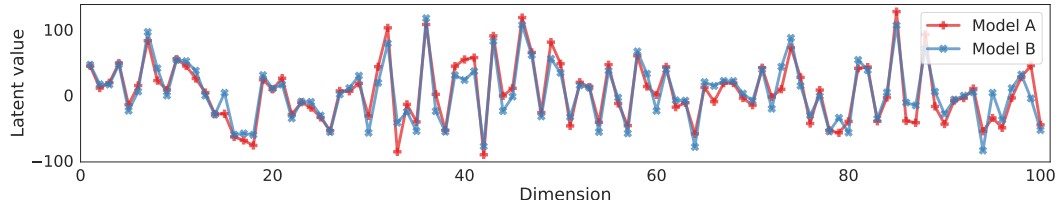

Figure 7: Comparing the first 100 dimensions of the latent code obtained for a random CIFAR-10 image. "Model A" and "Model B" are separately trained with different architectures.

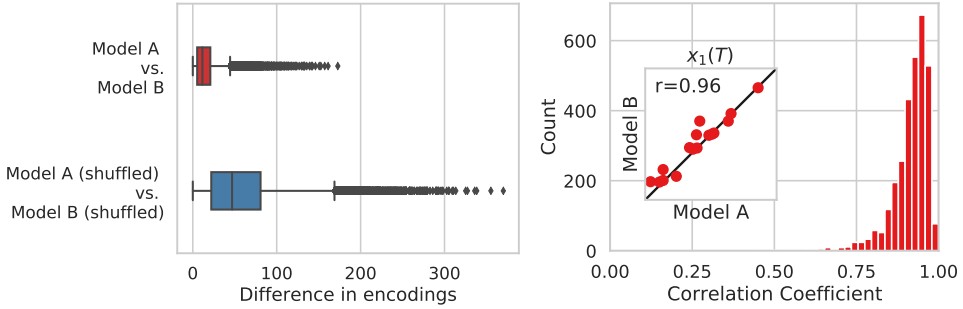

Figure 8: *Left*: The dimension-wise difference between encodings obtained by Model A and B. As a baseline, we also report the difference between shuffled representations of these two models. *Right*: The dimension-wise correlation coefficients of encodings obtained by Model A and Model B.

because

$$
\mathbf{x}_i = \frac{1}{\sqrt{1-\beta_{i+1}}}\left(\mathbf{x}_{i+1} + \beta_{i+1}\mathbf{s}_{\boldsymbol{\theta}*}(\mathbf{x}_{i+1}, i+1)\right) + \sqrt{\beta_{i+1}}\mathbf{z}_{i+1}
$$

$$
= \left(1 + \frac{1}{2}\beta_{i+1} + o(\beta_{i+1})\right)\left(\mathbf{x}_{i+1} + \beta_{i+1}\mathbf{s}_{\boldsymbol{\theta}*}(\mathbf{x}_{i+1}, i+1)\right) + \sqrt{\beta_{i+1}}\mathbf{z}_{i+1}
$$

$$
\approx \left(1 + \frac{1}{2}\beta_{i+1}\right)\left(\mathbf{x}_{i+1} + \beta_{i+1}\mathbf{s}_{\boldsymbol{\theta}*}(\mathbf{x}_{i+1}, i+1)\right) + \sqrt{\beta_{i+1}}\mathbf{z}_{i+1}
$$

$$
= \left(1 + \frac{1}{2}\beta_{i+1}\right)\mathbf{x}_{i+1} + \beta_{i+1}\mathbf{s}_{\boldsymbol{\theta}*}(\mathbf{x}_{i+1}, i+1) + \frac{1}{2}\beta_{i+1}^2\mathbf{s}_{\boldsymbol{\theta}*}(\mathbf{x}_{i+1}, i+1) + \sqrt{\beta_{i+1}}\mathbf{z}_{i+1}
$$

$$
\approx \left(1 + \frac{1}{2}\beta_{i+1}\right)\mathbf{x}_{i+1} + \beta_{i+1}\mathbf{s}_{\boldsymbol{\theta}*}(\mathbf{x}_{i+1}, i+1) + \sqrt{\beta_{i+1}}\mathbf{z}_{i+1}
$$

$$
= \left[2 - \left(1 - \frac{1}{2}\beta_{i+1}\right)\right]\mathbf{x}_{i+1} + \beta_{i+1}\mathbf{s}_{\boldsymbol{\theta}*}(\mathbf{x}_{i+1}, i+1) + \sqrt{\beta_{i+1}}\mathbf{z}_{i+1}
$$

$$
\approx \left[2 - \left(1 - \frac{1}{2}\beta_{i+1}\right) + o(\beta_{i+1})\right]\mathbf{x}_{i+1} + \beta_{i+1}\mathbf{s}_{\boldsymbol{\theta}*}(\mathbf{x}_{i+1}, i+1) + \sqrt{\beta_{i+1}}\mathbf{z}_{i+1}
$$

$$
= (2 - \sqrt{1-\beta_{i+1}})\mathbf{x}_{i+1} + \beta_{i+1}\mathbf{s}_{\boldsymbol{\theta}*}(\mathbf{x}_{i+1}, i+1) + \sqrt{\beta_{i+1}}\mathbf{z}_{i+1}.
$$

Therefore, the original ancestral sampler of Eq. (4) is essentially a different discretization to the same reverse-time SDE. This unifies the sampling method in Ho et al. (2020) as a numerical solver to the reverse-time VP SDE in our continuous framework.

## F    ANCESTRAL SAMPLING FOR SMLD MODELS

The ancestral sampling method for DDPM models can also be adapted to SMLD models. Consider a sequence of noise scales $\sigma_1 < \sigma_2 < \cdots < \sigma_N$ as in SMLD. By perturbing a data point $\mathbf{x}_0$ with these noise scales sequentially, we obtain a Markov chain $\mathbf{x}_0 \to \mathbf{x}_1 \to \cdots \to \mathbf{x}_N$, where

$$
p(\mathbf{x}_i \mid \mathbf{x}_{i-1}) = \mathcal{N}(\mathbf{x}_i; \mathbf{x}_{i-1}, (\sigma_i^2 - \sigma_{i-1}^2)\mathbf{I}), \quad i = 1, 2, \cdots, N.
$$

---

**Algorithm 1** Predictor-Corrector (PC) sampling

---

**Require:**
    $N$: Number of discretization steps for the reverse-time SDE
    $M$: Number of corrector steps
1: Initialize $\mathbf{x}_N \sim p_T(\mathbf{x})$
2: **for** $i = N - 1$ **to** 0 **do**
3:     $\mathbf{x}_i \leftarrow \text{Predictor}(\mathbf{x}_{i+1})$
4:     **for** $j = 1$ **to** $M$ **do**
5:         $\mathbf{x}_i \leftarrow \text{Corrector}(\mathbf{x}_i)$
6: **return** $\mathbf{x}_0$

---

Here we assume $\sigma_0 = 0$ to simplify notations. Following Ho et al. (2020), we can compute

$$q(\mathbf{x}_{i-1} \mid \mathbf{x}_i, \mathbf{x}_0) = \mathcal{N}\left(\mathbf{x}_{i-1}; \frac{\sigma_{i-1}^2}{\sigma_i^2}\mathbf{x}_i + \left(1 - \frac{\sigma_{i-1}^2}{\sigma_i^2}\right)\mathbf{x}_0, \frac{\sigma_{i-1}^2(\sigma_i^2 - \sigma_{i-1}^2)}{\sigma_i^2}\mathbf{I}\right).$$

If we parameterize the reverse transition kernel as $p_{\boldsymbol{\theta}}(\mathbf{x}_{i-1} \mid \mathbf{x}_i) = \mathcal{N}(\mathbf{x}_{i-1}; \boldsymbol{\mu}_{\boldsymbol{\theta}}(\mathbf{x}_i, i), \tau_i^2\mathbf{I})$, then

$$L_{t-1} = \mathbb{E}_q[D_{\mathrm{KL}}(q(\mathbf{x}_{i-1} \mid \mathbf{x}_i, \mathbf{x}_0) \,\|\, p_{\boldsymbol{\theta}}(\mathbf{x}_{i-1} \mid \mathbf{x}_i))]$$

$$= \mathbb{E}_q\left[\frac{1}{2\tau_i^2}\left\|\frac{\sigma_{i-1}^2}{\sigma_i^2}\mathbf{x}_i + \left(1 - \frac{\sigma_{i-1}^2}{\sigma_i^2}\right)\mathbf{x}_0 - \boldsymbol{\mu}_{\boldsymbol{\theta}}(\mathbf{x}_i, i)\right\|_2^2\right] + C$$

$$= \mathbb{E}_{\mathbf{x}_0, \mathbf{z}}\left[\frac{1}{2\tau_i^2}\left\|\mathbf{x}_i(\mathbf{x}_0, \mathbf{z}) - \frac{\sigma_i^2 - \sigma_{i-1}^2}{\sigma_i}\mathbf{z} - \boldsymbol{\mu}_{\boldsymbol{\theta}}(\mathbf{x}_i(\mathbf{x}_0, \mathbf{z}), i)\right\|_2^2\right] + C,$$

where $L_{t-1}$ is one representative term in the ELBO objective (see Eq. (8) in Ho et al. (2020)), $C$ is a constant that does not depend on $\boldsymbol{\theta}$, $\mathbf{z} \sim \mathcal{N}(\mathbf{0}, \mathbf{I})$, and $\mathbf{x}_i(\mathbf{x}_0, \mathbf{z}) = \mathbf{x}_0 + \sigma_i\mathbf{z}$. We can therefore parameterize $\boldsymbol{\mu}_{\boldsymbol{\theta}}(\mathbf{x}_i, i)$ via

$$\boldsymbol{\mu}_{\boldsymbol{\theta}}(\mathbf{x}_i, i) = \mathbf{x}_i + (\sigma_i^2 - \sigma_{i-1}^2)\mathbf{s}_{\boldsymbol{\theta}}(\mathbf{x}_i, i),$$

where $\mathbf{s}_{\boldsymbol{\theta}}(\mathbf{x}_i, i)$ is to estimate $\mathbf{z}/\sigma_i$. As in Ho et al. (2020), we let $\tau_i = \sqrt{\frac{\sigma_{i-1}^2(\sigma_i^2 - \sigma_{i-1}^2)}{\sigma_i^2}}$. Through ancestral sampling on $\prod_{i=1}^N p_{\boldsymbol{\theta}}(\mathbf{x}_{i-1} \mid \mathbf{x}_i)$, we obtain the following iteration rule

$$\mathbf{x}_{i-1} = \mathbf{x}_i + (\sigma_i^2 - \sigma_{i-1}^2)\mathbf{s}_{\boldsymbol{\theta}*}(\mathbf{x}_i, i) + \sqrt{\frac{\sigma_{i-1}^2(\sigma_i^2 - \sigma_{i-1}^2)}{\sigma_i^2}}\mathbf{z}_i, \quad i = 1, 2, \cdots, N, \qquad (47)$$

where $\mathbf{x}_N \sim \mathcal{N}(\mathbf{0}, \sigma_N^2\mathbf{I})$, $\boldsymbol{\theta}^*$ denotes the optimal parameter of $\mathbf{s}_{\boldsymbol{\theta}}$, and $\mathbf{z}_i \sim \mathcal{N}(\mathbf{0}, \mathbf{I})$. We call Eq. (47) the ancestral sampling method for SMLD models.

## G   PREDICTOR-CORRECTOR SAMPLERS

**Predictor-Corrector (PC) sampling**   The predictor can be any numerical solver for the reverse-time SDE with a fixed discretization strategy. The corrector can be any score-based MCMC approach. In PC sampling, we alternate between the predictor and corrector, as described in Algorithm 1. For example, when using the reverse diffusion SDE solver (Appendix E) as the predictor, and annealed Langevin dynamics (Song & Ermon, 2019) as the corrector, we have Algorithms 2 and 3 for VE and VP SDEs respectively, where $\{\epsilon_i\}_{i=0}^{N-1}$ are step sizes for Langevin dynamics as specified below.

**The corrector algorithms**   We take the schedule of annealed Langevin dynamics in Song & Ermon (2019), but re-frame it with slight modifications in order to get better interpretability and empirical performance. We provide the corrector algorithms in Algorithms 4 and 5 respectively, where we call $r$ the "signal-to-noise" ratio. We determine the step size $\epsilon$ using the norm of the Gaussian noise $\|\mathbf{z}\|_2$, norm of the score-based model $\|\mathbf{s}_{\boldsymbol{\theta}*}\|_2$ and the signal-to-noise ratio $r$. When sampling a large batch of samples together, we replace the norm $\|\cdot\|_2$ with the average norm across the mini-batch. When the batch size is small, we suggest replacing $\|\mathbf{z}\|_2$ with $\sqrt{d}$, where $d$ is the dimensionality of $\mathbf{z}$.

| **Algorithm 2** PC sampling (VE SDE) | **Algorithm 3** PC sampling (VP SDE) |
|---|---|
| 1: $\mathbf{x}_N \sim \mathcal{N}(\mathbf{0}, \sigma_{\max}^2 \mathbf{I})$ | 1: $\mathbf{x}_N \sim \mathcal{N}(\mathbf{0}, \mathbf{I})$ |
| 2: **for** $i = N-1$ **to** $0$ **do** | 2: **for** $i = N-1$ **to** $0$ **do** |
| 3: $\quad \mathbf{x}_i' \leftarrow \mathbf{x}_{i+1} + (\sigma_{i+1}^2 - \sigma_i^2)\mathbf{s}_{\boldsymbol{\theta}*}(\mathbf{x}_{i+1}, \sigma_{i+1})$ | 3: $\quad \mathbf{x}_i' \leftarrow (2 - \sqrt{1 - \beta_{i+1}})\mathbf{x}_{i+1} + \beta_{i+1}\mathbf{s}_{\boldsymbol{\theta}*}(\mathbf{x}_{i+1}, i+1)$ |
| 4: $\quad \mathbf{z} \sim \mathcal{N}(\mathbf{0}, \mathbf{I})$ | 4: $\quad \mathbf{z} \sim \mathcal{N}(\mathbf{0}, \mathbf{I})$ |
| 5: $\quad \mathbf{x}_i \leftarrow \mathbf{x}_i' + \sqrt{\sigma_{i+1}^2 - \sigma_i^2}\mathbf{z}$ | 5: $\quad \mathbf{x}_i \leftarrow \mathbf{x}_i' + \sqrt{\beta_{i+1}}\mathbf{z}$ |
| 6: $\quad$ **for** $j = 1$ **to** $M$ **do** | 6: $\quad$ **for** $j = 1$ **to** $M$ **do** |
| 7: $\quad\quad \mathbf{z} \sim \mathcal{N}(\mathbf{0}, \mathbf{I})$ | 7: $\quad\quad \mathbf{z} \sim \mathcal{N}(\mathbf{0}, \mathbf{I})$ |
| 8: $\quad\quad \mathbf{x}_i \leftarrow \mathbf{x}_i + \epsilon_i \mathbf{s}_{\boldsymbol{\theta}*}(\mathbf{x}_i, \sigma_i) + \sqrt{2\epsilon_i}\mathbf{z}$ | 8: $\quad\quad \mathbf{x}_i \leftarrow \mathbf{x}_i + \epsilon_i \mathbf{s}_{\boldsymbol{\theta}*}(\mathbf{x}_i, i) + \sqrt{2\epsilon_i}\mathbf{z}$ |
| 9: **return** $\mathbf{x}_0$ | 9: **return** $\mathbf{x}_0$ |

Predictor

Corrector

| **Algorithm 4** Corrector algorithm (VE SDE). | **Algorithm 5** Corrector algorithm (VP SDE). |
|---|---|
| **Require:** $\{\sigma_i\}_{i=1}^N, r, N, M.$ | **Require:** $\{\beta_i\}_{i=1}^N, \{\alpha_i\}_{i=1}^N, r, N, M.$ |
| 1: $\mathbf{x}_N^0 \sim \mathcal{N}(\mathbf{0}, \sigma_{\max}^2 \mathbf{I})$ | 1: $\mathbf{x}_N^0 \sim \mathcal{N}(\mathbf{0}, \mathbf{I})$ |
| 2: **for** $i \leftarrow N$ **to** $1$ **do** | 2: **for** $i \leftarrow N$ **to** $1$ **do** |
| 3: $\quad$ **for** $j \leftarrow 1$ **to** $M$ **do** | 3: $\quad$ **for** $j \leftarrow 1$ **to** $M$ **do** |
| 4: $\quad\quad \mathbf{z} \sim \mathcal{N}(\mathbf{0}, \mathbf{I})$ | 4: $\quad\quad \mathbf{z} \sim \mathcal{N}(\mathbf{0}, \mathbf{I})$ |
| 5: $\quad\quad \mathbf{g} \leftarrow \mathbf{s}_{\boldsymbol{\theta}*}(\mathbf{x}_i^{j-1}, \sigma_i)$ | 5: $\quad\quad \mathbf{g} \leftarrow \mathbf{s}_{\boldsymbol{\theta}*}(\mathbf{x}_i^{j-1}, i)$ |
| 6: $\quad\quad \epsilon \leftarrow 2(r\|\mathbf{z}\|_2 / \|\mathbf{g}\|_2)^2$ | 6: $\quad\quad \epsilon \leftarrow 2\alpha_i(r\|\mathbf{z}\|_2 / \|\mathbf{g}\|_2)^2$ |
| 7: $\quad\quad \mathbf{x}_i^j \leftarrow \mathbf{x}_i^{j-1} + \epsilon \mathbf{g} + \sqrt{2\epsilon}\,\mathbf{z}$ | 7: $\quad\quad \mathbf{x}_i^j \leftarrow \mathbf{x}_i^{j-1} + \epsilon \mathbf{g} + \sqrt{2\epsilon}\,\mathbf{z}$ |
| 8: $\quad \mathbf{x}_{i-1}^0 \leftarrow \mathbf{x}_i^M$ | 8: $\quad \mathbf{x}_{i-1}^0 \leftarrow \mathbf{x}_i^M$ |
| $\quad$ **return** $\mathbf{x}_0^0$ | $\quad$ **return** $\mathbf{x}_0^0$ |

**Denoising** For both SMLD and DDPM models, the generated samples typically contain small noise that is hard to detect by humans. As noted by Jolicoeur-Martineau et al. (2020), FIDs can be significantly worse without removing this noise. This unfortunate sensitivity to noise is also part of the reason why NCSN models trained with SMLD has been performing worse than DDPM models in terms of FID, because the former does not use a denoising step at the end of sampling, while the latter does. In all experiments of this paper we ensure there is a single denoising step at the end of sampling, using Tweedie's formula (Efron, 2011).

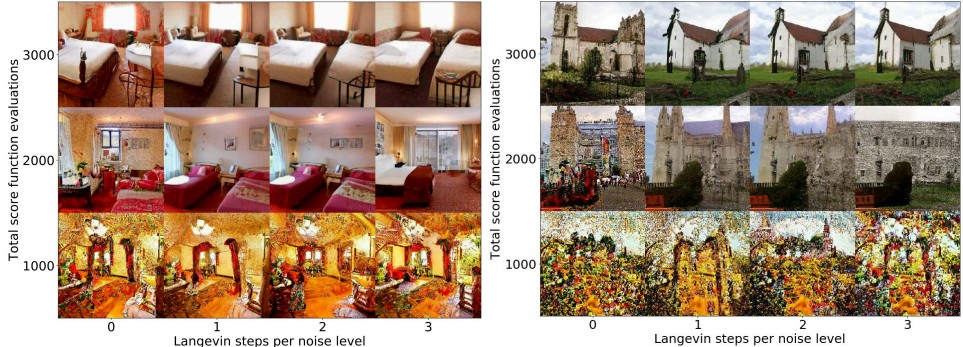

Figure 9: PC sampling for LSUN bedroom and church. The vertical axis corresponds to the total computation, and the horizontal axis represents the amount of computation allocated to the corrector. Samples are the best when computation is split between the predictor and corrector.

**Training** We use the same architecture in Ho et al. (2020) for our score-based models. For the VE SDE, we train a model with the original SMLD objective in Eq. (1); similarly for the VP SDE, we use the original DDPM objective in Eq. (3). We apply a total number of 1000 noise scales for training both models. For results in Fig. 9, we train an NCSN++ model (definition in Appendix H) on

Table 4: Comparing different samplers on CIFAR-10, where "P2000" uses the rounding interpolation between noise scales. Shaded regions are obtained with the same computation (number of score function evaluations). Mean and standard deviation are reported over five sampling runs.

| FID↓   Sampler \newline\newline Predictor | Variance Exploding SDE (SMLD) | | | | Variance Preserving SDE (DDPM) | | | |
|---|---|---|---|---|---|---|---|---|
| | P1000 | P2000 | C2000 | PC1000 | P1000 | P2000 | C2000 | PC1000 |
| ancestral sampling | $4.98 \pm .06$ | $4.92 \pm .02$ | | $\mathbf{3.62} \pm \mathbf{.03}$ | $3.24 \pm .02$ | $\mathbf{3.11} \pm \mathbf{.03}$ | | $3.21 \pm .02$ |
| reverse diffusion | $4.79 \pm .07$ | $4.72 \pm .07$ | $20.43 \pm .07$ | $\mathbf{3.60} \pm \mathbf{.02}$ | $3.21 \pm .02$ | $\mathbf{3.10} \pm \mathbf{.03}$ | $19.06 \pm .06$ | $3.18 \pm .01$ |
| probability flow | $15.41 \pm .15$ | $12.87 \pm .09$ | | $\mathbf{3.51} \pm \mathbf{.04}$ | $3.59 \pm .04$ | $3.25 \pm .04$ | | $\mathbf{3.06} \pm \mathbf{.03}$ |

Table 5: Optimal signal-to-noise ratios of different samplers. "P1000" or "P2000": predictor-only samplers using 1000 or 2000 steps. "C2000": corrector-only samplers using 2000 steps. "PC1000": PC samplers using 1000 predictor and 1000 corrector steps.

| $r$   Sampler \newline\newline Predictor | VE SDE (SMLD) | | | | VP SDE (DDPM) | | | |
|---|---|---|---|---|---|---|---|---|
| | P1000 | P2000 | C2000 | PC1000 | P1000 | P2000 | C2000 | PC1000 |
| ancestral sampling | - | - | | 0.17 | - | - | | 0.01 |
| reverse diffusion | - | - | 0.22 | 0.16 | - | - | 0.27 | 0.01 |
| probability flow | - | - | | 0.17 | - | - | | 0.04 |

$256 \times 256$ LSUN bedroom and church_outdoor (Yu et al., 2015) datasets with the VE SDE and our continuous objective Eq. (7). The batch size is fixed to 128 on CIFAR-10 and 64 on LSUN.

**Ad-hoc interpolation methods for noise scales**   Models in this experiment are all trained with 1000 noise scales. To get results for P2000 (predictor-only sampler using 2000 steps) which requires 2000 noise scales, we need to interpolate between 1000 noise scales at test time. The specific architecture of the noise-conditional score-based model in Ho et al. (2020) uses sinusoidal positional embeddings for conditioning on integer time steps. This allows us to interpolate between noise scales at test time in an ad-hoc way (while it is hard to do so for other architectures like the one in Song & Ermon (2019)). Specifically, for SMLD models, we keep $\sigma_{\min}$ and $\sigma_{\max}$ fixed and double the number of time steps. For DDPM models, we halve $\beta_{\min}$ and $\beta_{\max}$ before doubling the number of time steps. Suppose $\{\mathbf{s}_{\boldsymbol{\theta}}(\mathbf{x}, i)\}_{i=0}^{N-1}$ is a score-based model trained on $N$ time steps, and let $\{\mathbf{s}'_{\boldsymbol{\theta}}(\mathbf{x}, i)\}_{i=0}^{2N-1}$ denote the corresponding interpolated score-based model at $2N$ time steps. We test two different interpolation strategies for time steps: linear interpolation where $\mathbf{s}'_{\boldsymbol{\theta}}(\mathbf{x}, i) = \mathbf{s}_{\boldsymbol{\theta}}(\mathbf{x}, i/2)$ and rounding interpolation where $\mathbf{s}'_{\boldsymbol{\theta}}(\mathbf{x}, i) = \mathbf{s}_{\boldsymbol{\theta}}(\mathbf{x}, \lfloor i/2 \rfloor)$. We provide results with linear interpolation in Table 1, and give results of rounding interpolation in Table 4. We observe that different interpolation methods result in performance differences but maintain the general trend of predictor-corrector methods performing on par or better than predictor-only or corrector-only samplers.

**Hyper-parameters of the samplers**   For Predictor-Corrector and corrector-only samplers on CIFAR-10, we search for the best signal-to-noise ratio ($r$) over a grid that increments at 0.01. We report the best $r$ in Table 5. For LSUN bedroom/church_outdoor, we fix $r$ to 0.075. Unless otherwise noted, we use one corrector step per noise scale for all PC samplers. We use two corrector steps per noise scale for corrector-only samplers on CIFAR-10. For sample generation, the batch size is 1024 on CIFAR-10 and 8 on LSUN bedroom/church_outdoor.

# H   ARCHITECTURE IMPROVEMENTS

We explored several architecture designs to improve score-based models for both VE and VP SDEs. Our endeavor gives rise to new state-of-the-art sample quality on CIFAR-10, new state-of-the-art likelihood on uniformly dequantized CIFAR-10, and enables the first high-fidelity image samples of resolution $1024 \times 1024$ from score-based generative models. Code and checkpoints are open-sourced at https://github.com/yang-song/score_sde.

## H.1 SETTINGS FOR ARCHITECTURE EXPLORATION

Unless otherwise noted, all models are trained for 1.3M iterations, and we save one checkpoint per 50k iterations. For VE SDEs, we consider two datasets: $32 \times 32$ CIFAR-10 (Krizhevsky et al., 2009) and $64 \times 64$ CelebA (Liu et al., 2015), pre-processed following Song & Ermon (2020). We compare different configurations based on their FID scores averaged over checkpoints after 0.5M iterations. For VP SDEs, we only consider the CIFAR-10 dataset to save computation, and compare models based on the average FID scores over checkpoints obtained between 0.25M and 0.5M iterations, because FIDs turn to increase after 0.5M iterations for VP SDEs.

All FIDs are computed on 50k samples with `tensorflow_gan`. For sampling, we use the PC sampler discretized at 1000 time steps. We choose reverse diffusion (see Appendix E) as the predictor. We use one corrector step per update of the predictor for VE SDEs with a signal-to-noise ratio of 0.16, but save the corrector step for VP SDEs since correctors there only give slightly better results but require double computation. We follow Ho et al. (2020) for optimization, including the learning rate, gradient clipping, and learning rate warm-up schedules. Unless otherwise noted, models are trained with the original discrete SMLD and DDPM objectives in Eqs. (1) and (3) and use a batch size of 128. The optimal architectures found under these settings are subsequently transferred to continuous objectives and deeper models. We also directly transfer the best architecture for VP SDEs to sub-VP SDEs, given the similarity of these two SDEs.

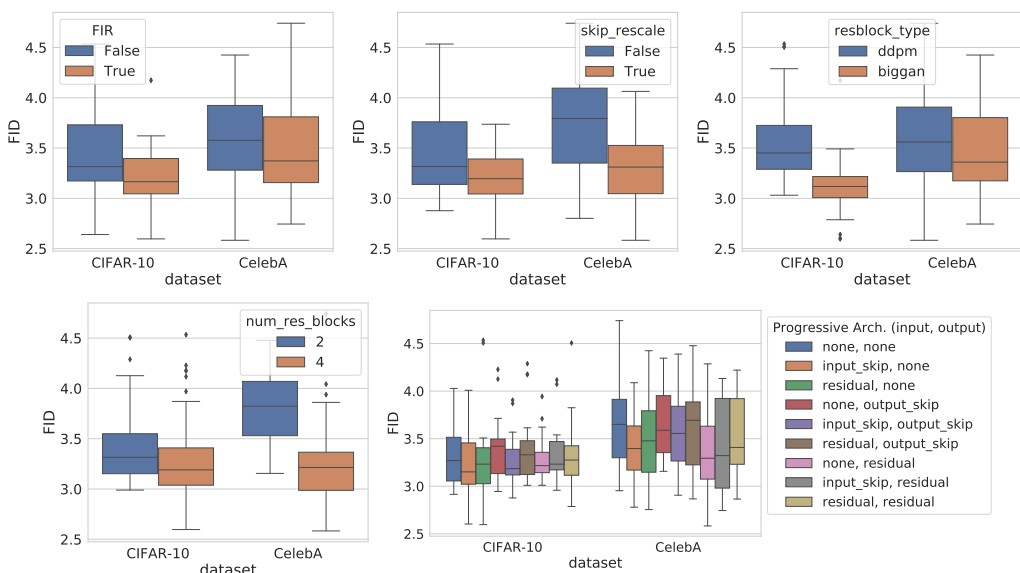

Figure 10: The effects of different architecture components for score-based models trained with VE perturbations.

Our architecture is mostly based on Ho et al. (2020). We additionally introduce the following components to maximize the potential improvement of score-based models.

1. Upsampling and downsampling images with anti-aliasing based on Finite Impulse Response (FIR) (Zhang, 2019). We follow the same implementation and hyper-parameters in StyleGAN-2 (Karras et al., 2020b).

2. Rescaling all skip connections by $1/\sqrt{2}$. This has been demonstrated effective in several best-in-class GAN models, including ProgressiveGAN (Karras et al., 2018), StyleGAN (Karras et al., 2019) and StyleGAN-2 (Karras et al., 2020b).

3. Replacing the original residual blocks in DDPM with residual blocks from BigGAN (Brock et al., 2018).

4. Increasing the number of residual blocks per resolution from 2 to 4.

5. Incorporating progressive growing architectures. We consider two progressive architectures for input: "input skip" and "residual", and two progressive architectures for output: "output skip" and "residual". These progressive architectures are defined and implemented according to StyleGAN-2.

We also tested equalized learning rates, a trick used in very successful models like Progressive-GAN (Karras et al., 2018) and StyleGAN (Karras et al., 2019). However, we found it harmful at an early stage of our experiments, and therefore decided not to explore more on it.

The exponential moving average (EMA) rate has a significant impact on performance. For models trained with VE perturbations, we notice that 0.999 works better than 0.9999, whereas for models trained with VP perturbations it is the opposite. We therefore use an EMA rate of 0.999 and 0.9999 for VE and VP models respectively.

## H.2 Results on CIFAR-10

All architecture components introduced above can improve the performance of score-based models trained with VE SDEs, as shown in Fig. 10. The box plots demonstrate the importance of each component when other components can vary freely. On both CIFAR-10 and CelebA, the additional components that we explored always improve the performance on average for VE SDEs. For progressive growing, it is not clear which combination of configurations consistently performs the best, but the results are typically better than when no progressive growing architecture is used. Our best score-based model for VE SDEs 1) uses FIR upsampling/downsampling, 2) rescales skip connections, 3) employs BigGAN-type residual blocks, 4) uses 4 residual blocks per resolution instead of 2, and 5) uses "residual" for input and no progressive growing architecture for output. We name this model "NCSN++", following the naming convention of previous SMLD models (Song & Ermon, 2019; 2020).

We followed a similar procedure to examine these architecture components for VP SDEs, except that we skipped experiments on CelebA due to limited computing resources. The NCSN++ architecture worked decently well for VP SDEs, ranked 4th place over all 144 possible configurations. The top configuration, however, has a slightly different structure, which uses no FIR upsampling/downsampling and no progressive growing architecture compared to NCSN++. We name this model "DDPM++", following the naming convention of Ho et al. (2020).

The basic NCSN++ model with 4 residual blocks per resolution achieves an FID of 2.45 on CIFAR-10, whereas the basic DDPM++ model achieves an FID of 2.78. Here in order to match the convention used in Karras et al. (2018); Song & Ermon (2019) and Ho et al. (2020), we report the lowest FID value over the course of training, rather than the average FID value over checkpoints after 0.5M iterations (used for comparing different models of VE SDEs) or between 0.25M and 0.5M iterations (used for comparing VP SDE models) in our architecture exploration.

Switching from discrete training objectives to continuous ones in Eq. (7) further improves the FID values for all SDEs. To condition the NCSN++ model on continuous time variables, we change positional embeddings, the layers in Ho et al. (2020) for conditioning on discrete time steps, to random Fourier feature embeddings (Tancik et al., 2020). The scale parameter of these random Fourier feature embeddings is fixed to 16. We also reduce the number of training iterations to 0.95M to suppress overfitting. These changes improve the FID on CIFAR-10 from 2.45 to 2.38 for NCSN++ trained with the VE SDE, resulting in a model called "NCSN++ cont.". In addition, we can further improve the FID from 2.38 to 2.20 by doubling the number of residual blocks per resolution for NCSN++ cont., resulting in the model denoted as "NCSN++ cont. (deep)". All quantitative results are summarized in Table 3, and we provide random samples from our best model in Fig. 11.

Similarly, we can also condition the DDPM++ model on continuous time steps, resulting in a model "DDPM++ cont.". When trained with the VP SDE, it improves the FID of 2.78 from DDPM++ to 2.55. When trained with the sub-VP SDE, it achieves an FID of 2.61. To get better performance, we used the Euler-Maruyama solver as the predictor for continuously-trained models, instead of the ancestral sampling predictor or the reverse diffusion predictor. This is because the discretization strategy of the original DDPM method does not match the variance of the continuous process well when $t \to 0$, which significantly hurts FID scores. As shown in Table 2, the likelihood values are 3.21 and 3.05 bits/dim for VP and sub-VP SDEs respectively. Doubling the depth, and trainin with

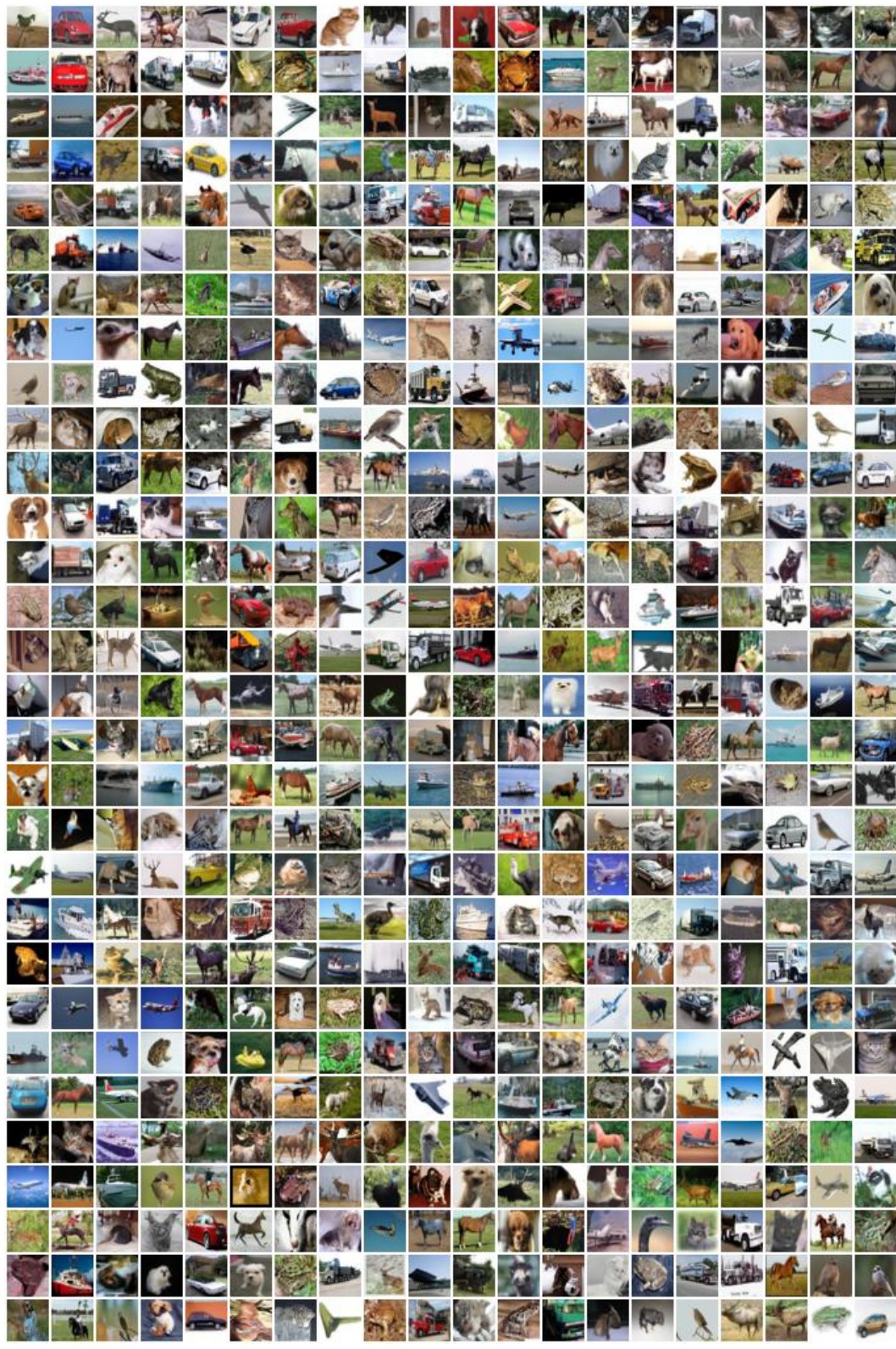

Figure 11: Unconditional CIFAR-10 samples from NCSN++ cont. (deep, VE).

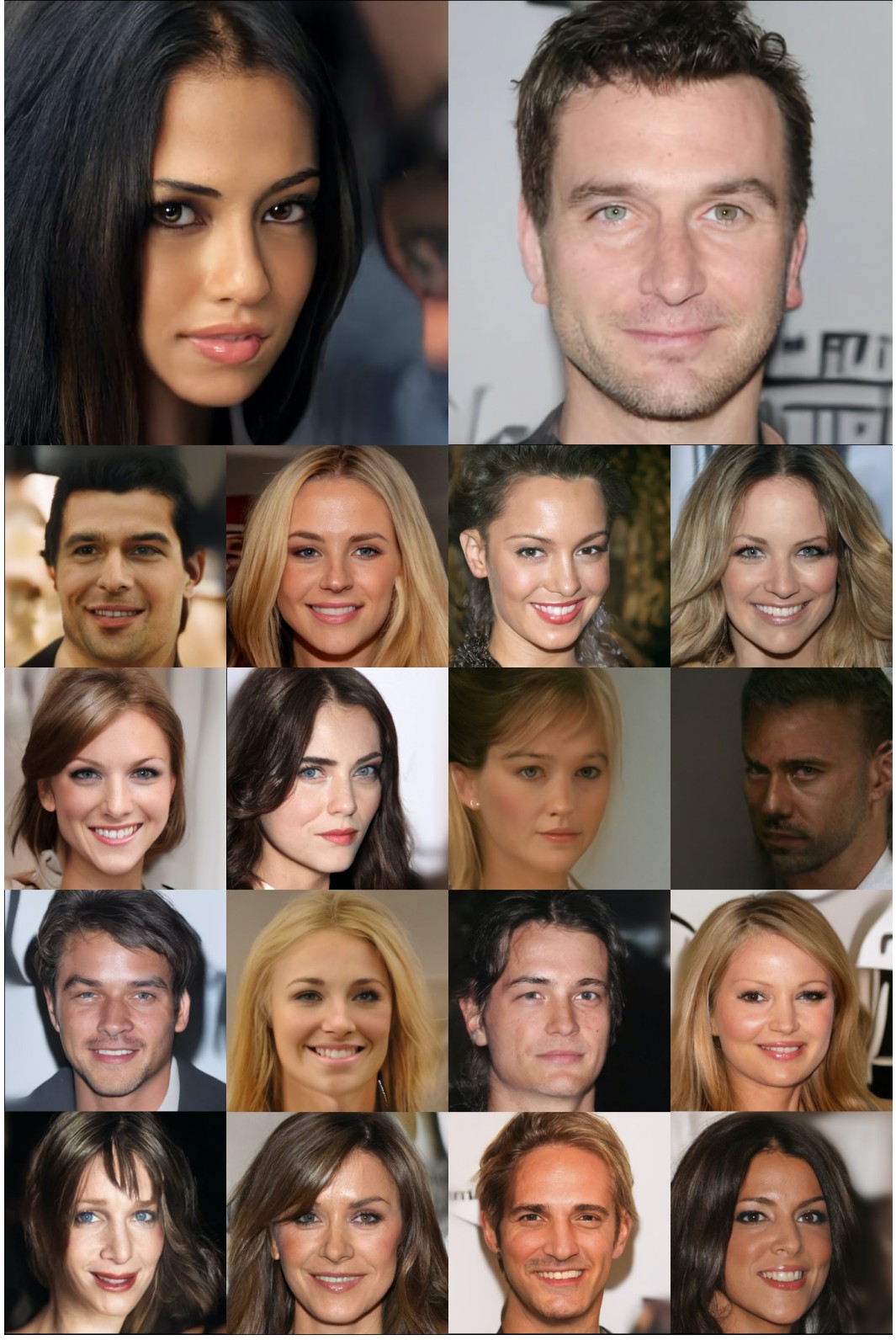

Figure 12: Samples on $1024 \times 1024$ CelebA-HQ from a modified NCSN++ model trained with the VE SDE.

0.95M iterations, we can improve both FID and bits/dim for both VP and sub-VP SDEs, leading to a model "DDPM++ cont. (deep)". Its FID score is 2.41, same for both VP and sub-VP SDEs. When trained with the sub-VP SDE, it can achieve a likelihood of 2.99 bits/dim. Here all likelihood values are reported for the last checkpoint during training.

### H.3 HIGH RESOLUTION IMAGES

Encouraged by the success of NCSN++ on CIFAR-10, we proceed to test it on $1024 \times 1024$ CelebA-HQ (Karras et al., 2018), a task that was previously only achievable by some GAN models and VQ-VAE-2 (Razavi et al., 2019). We used a batch size of 8, increased the EMA rate to 0.9999, and trained a model similar to NCSN++ with the continuous objective (Eq. (7)) for around 2.4M iterations (please find the detailed architecture in our code release.) We use the PC sampler discretized at 2000 steps with the reverse diffusion predictor, one Langevin step per predictor update and a signal-to-noise ratio of 0.15. The scale parameter for the random Fourier feature embeddings is fixed to 16. We use the "input skip" progressive architecture for the input, and "output skip" progressive architecture for the output. We provide samples in Fig. 12. Although these samples are not perfect (*e.g.*, there are visible flaws on facial symmetry), we believe these results are encouraging and can demonstrate the scalability of our approach. Future work on more effective architectures are likely to significantly advance the performance of score-based generative models on this task.

## I CONTROLLABLE GENERATION

Consider a forward SDE with the following general form

$$\mathrm{d}\mathbf{x} = \mathbf{f}(\mathbf{x}, t)\mathrm{d}t + \mathbf{G}(\mathbf{x}, t)\mathrm{d}\mathbf{w},$$

and suppose the initial state distribution is $p_0(\mathbf{x}(0) \mid \mathbf{y})$. The density at time $t$ is $p_t(\mathbf{x}(t) \mid \mathbf{y})$ when conditioned on $\mathbf{y}$. Therefore, using Anderson (1982), the reverse-time SDE is given by

$$\mathrm{d}\mathbf{x} = \{\mathbf{f}(\mathbf{x}, t) - \nabla \cdot [\mathbf{G}(\mathbf{x}, t)\mathbf{G}(\mathbf{x}, t)^\mathsf{T}] - \mathbf{G}(\mathbf{x}, t)\mathbf{G}(\mathbf{x}, t)^\mathsf{T}\nabla_\mathbf{x} \log p_t(\mathbf{x} \mid \mathbf{y})\}\mathrm{d}t + \mathbf{G}(\mathbf{x}, t)\mathrm{d}\bar{\mathbf{w}}. \quad (48)$$

Since $p_t(\mathbf{x}(t) \mid \mathbf{y}) \propto p_t(\mathbf{x}(t))p(\mathbf{y} \mid \mathbf{x}(t))$, the score $\nabla_\mathbf{x} \log p_t(\mathbf{x} \mid \mathbf{y})$ can be computed easily by

$$\nabla_\mathbf{x} \log p_t(\mathbf{x}(t) \mid \mathbf{y}) = \nabla_\mathbf{x} \log p_t(\mathbf{x}(t)) + \nabla_\mathbf{x} \log p(\mathbf{y} \mid \mathbf{x}(t)). \quad (49)$$

This subsumes the conditional reverse-time SDE in Eq. (14) as a special case. All sampling methods we have discussed so far can be applied to the conditional reverse-time SDE for sample generation.

### I.1 CLASS-CONDITIONAL SAMPLING

When $\mathbf{y}$ represents class labels, we can train a time-dependent classifier $p_t(\mathbf{y} \mid \mathbf{x}(t))$ for class-conditional sampling. Since the forward SDE is tractable, we can easily create a pair of training data $(\mathbf{x}(t), \mathbf{y})$ by first sampling $(\mathbf{x}(0), \mathbf{y})$ from a dataset and then obtaining $\mathbf{x}(t) \sim p_{0t}(\mathbf{x}(t) \mid \mathbf{x}(0))$. Afterwards, we may employ a mixture of cross-entropy losses over different time steps, like Eq. (7), to train the time-dependent classifier $p_t(\mathbf{y} \mid \mathbf{x}(t))$.

To test this idea, we trained a Wide ResNet (Zagoruyko & Komodakis, 2016) (`Wide-ResNet-28-10`) on CIFAR-10 with VE perturbations. The classifier is conditioned on $\log \sigma_i$ using random Fourier features (Tancik et al., 2020), and the training objective is a simple sum of cross-entropy losses sampled at different scales. We provide a plot to show the accuracy of this classifier over noise scales in Fig. 13. The score-based model is an unconditional NCSN++ (4 blocks/resolution) in Table 3, and we generate samples using the PC algorithm with 2000 discretization steps. The class-conditional samples are provided in Fig. 4, and an extended set of conditional samples is given in Fig. 13.

### I.2 IMPUTATION

Imputation is a special case of conditional sampling. Denote by $\Omega(\mathbf{x})$ and $\bar{\Omega}(\mathbf{x})$ the known and unknown dimensions of $\mathbf{x}$ respectively, and let $\mathbf{f}_{\bar{\Omega}}(\cdot, t)$ and $\mathbf{G}_{\bar{\Omega}}(\cdot, t)$ denote $\mathbf{f}(\cdot, t)$ and $\mathbf{G}(\cdot, t)$ restricted to the unknown dimensions. For VE/VP SDEs, the drift coefficient $\mathbf{f}(\cdot, t)$ is element-wise, and the diffusion coefficient $\mathbf{G}(\cdot, t)$ is diagonal. When $\mathbf{f}(\cdot, t)$ is element-wise, $\mathbf{f}_{\bar{\Omega}}(\cdot, t)$ denotes the same

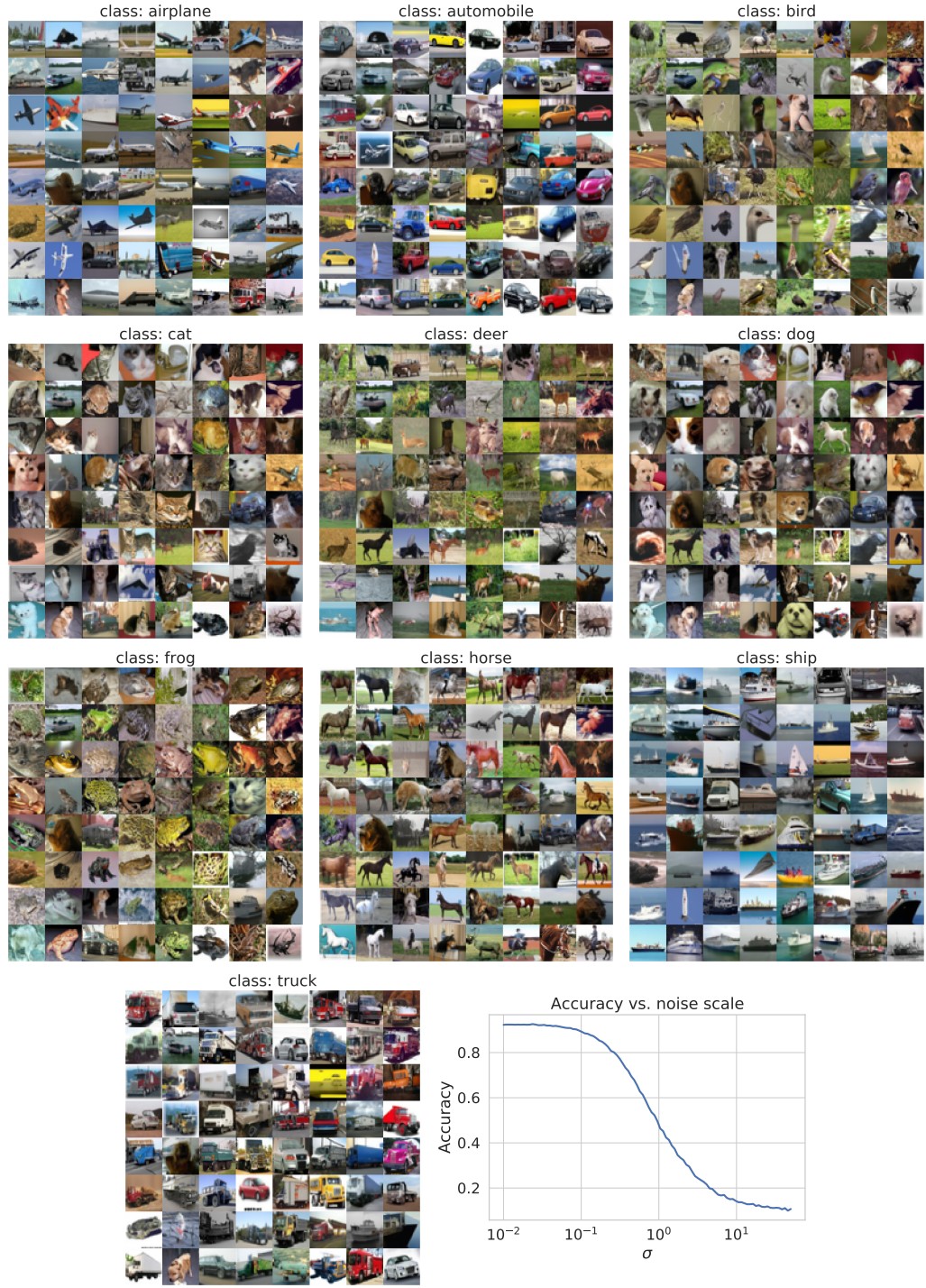

Figure 13: Class-conditional image generation by solving the conditional reverse-time SDE with PC. The curve shows the accuracy of our noise-conditional classifier over different noise scales.

element-wise function applied only to the unknown dimensions. When $\mathbf{G}(\cdot, t)$ is diagonal, $\mathbf{G}_{\bar{\Omega}}(\cdot, t)$ denotes the sub-matrix restricted to unknown dimensions.

For imputation, our goal is to sample from $p(\bar{\Omega}(\mathbf{x}(0)) \mid \Omega(\mathbf{x}(0)) = \mathbf{y})$. Define a new diffusion process $\mathbf{z}(t) = \bar{\Omega}(\mathbf{x}(t))$, and note that the SDE for $\mathbf{z}(t)$ can be written as

$$\mathrm{d}\mathbf{z} = \mathbf{f}_{\bar{\Omega}}(\mathbf{z}, t)\mathrm{d}t + \mathbf{G}_{\bar{\Omega}}(\mathbf{z}, t)\mathrm{d}\mathbf{w}.$$

The reverse-time SDE, conditioned on $\Omega(\mathbf{x}(0)) = \mathbf{y}$, is given by

$$\mathrm{d}\mathbf{z} = \big\{\mathbf{f}_{\bar{\Omega}}(\mathbf{z}, t) - \nabla \cdot [\mathbf{G}_{\bar{\Omega}}(\mathbf{z}, t)\mathbf{G}_{\bar{\Omega}}(\mathbf{z}, t)^{\mathsf{T}}]$$
$$- \mathbf{G}_{\bar{\Omega}}(\mathbf{z}, t)\mathbf{G}_{\bar{\Omega}}(\mathbf{z}, t)^{\mathsf{T}}\nabla_{\mathbf{z}} \log p_t(\mathbf{z} \mid \Omega(\mathbf{z}(0)) = \mathbf{y})\big\}\mathrm{d}t + \mathbf{G}_{\bar{\Omega}}(\mathbf{z}, t)\mathrm{d}\bar{\mathbf{w}}.$$

Although $p_t(\mathbf{z}(t) \mid \Omega(\mathbf{x}(0)) = \mathbf{y})$ is in general intractable, it can be approximated. Let $A$ denote the event $\Omega(\mathbf{x}(0)) = \mathbf{y}$. We have

$$p_t(\mathbf{z}(t) \mid \Omega(\mathbf{x}(0)) = \mathbf{y}) = p_t(\mathbf{z}(t) \mid A) = \int p_t(\mathbf{z}(t) \mid \Omega(\mathbf{x}(t)), A)p_t(\Omega(\mathbf{x}(t)) \mid A)\mathrm{d}\Omega(\mathbf{x}(t))$$
$$= \mathbb{E}_{p_t(\Omega(\mathbf{x}(t))|A)}[p_t(\mathbf{z}(t) \mid \Omega(\mathbf{x}(t)), A)]$$
$$\approx \mathbb{E}_{p_t(\Omega(\mathbf{x}(t))|A)}[p_t(\mathbf{z}(t) \mid \Omega(\mathbf{x}(t)))]$$
$$\approx p_t(\mathbf{z}(t) \mid \hat{\Omega}(\mathbf{x}(t))),$$

where $\hat{\Omega}(\mathbf{x}(t))$ is a random sample from $p_t(\Omega(\mathbf{x}(t)) \mid A)$, which is typically a tractable distribution. Therefore,

$$\nabla_{\mathbf{z}} \log p_t(\mathbf{z}(t) \mid \Omega(\mathbf{x}(0)) = \mathbf{y}) \approx \nabla_{\mathbf{z}} \log p_t(\mathbf{z}(t) \mid \hat{\Omega}(\mathbf{x}(t)))$$
$$= \nabla_{\mathbf{z}} \log p_t([\mathbf{z}(t); \hat{\Omega}(\mathbf{x}(t))]),$$

where $[\mathbf{z}(t); \hat{\Omega}(\mathbf{x}(t))]$ denotes a vector $\mathbf{u}(t)$ such that $\Omega(\mathbf{u}(t)) = \hat{\Omega}(\mathbf{x}(t))$ and $\bar{\Omega}(\mathbf{u}(t)) = \mathbf{z}(t)$, and the identity holds because $\nabla_{\mathbf{z}} \log p_t([\mathbf{z}(t); \hat{\Omega}(\mathbf{x}(t))]) = \nabla_{\mathbf{z}} \log p_t(\mathbf{z}(t) \mid \hat{\Omega}(\mathbf{x}(t))) + \nabla_{\mathbf{z}} \log p(\hat{\Omega}(\mathbf{x}(t))) = \nabla_{\mathbf{z}} \log p_t(\mathbf{z}(t) \mid \hat{\Omega}(\mathbf{x}(t)))$.

We provided an extended set of inpainting results in Figs. 14 and 15.

### I.3 COLORIZATION

Colorization is a special case of imputation, except that the known data dimensions are coupled. We can decouple these data dimensions by using an orthogonal linear transformation to map the gray-scale image to a separate channel in a different space, and then perform imputation to complete the other channels before transforming everything back to the original image space. The orthogonal matrix we used to decouple color channels is

$$\begin{pmatrix} 0.577 & -0.816 & 0 \\ 0.577 & 0.408 & 0.707 \\ 0.577 & 0.408 & -0.707 \end{pmatrix}.$$

Because the transformations are all orthogonal matrices, the standard Wiener process $\mathbf{w}(t)$ will still be a standard Wiener process in the transformed space, allowing us to build an SDE and use the same imputation method in Appendix I.2. We provide an extended set of colorization results in Figs. 16 and 17.

### I.4 SOLVING GENERAL INVERSE PROBLEMS

Suppose we have two random variables $\mathbf{x}$ and $\mathbf{y}$, and we know the forward process of generating $\mathbf{y}$ from $\mathbf{x}$, given by $p(\mathbf{y} \mid \mathbf{x})$. The inverse problem is to obtain $\mathbf{x}$ from $\mathbf{y}$, that is, generating samples from $p(\mathbf{x} \mid \mathbf{y})$. In principle, we can estimate the prior distribution $p(\mathbf{x})$ and obtain $p(\mathbf{x} \mid \mathbf{y})$ using Bayes' rule: $p(\mathbf{x} \mid \mathbf{y}) = p(\mathbf{x})p(\mathbf{y} \mid \mathbf{x})/p(\mathbf{y})$. In practice, however, both estimating the prior and performing Bayesian inference are non-trivial.

Leveraging Eq. (48), score-based generative models provide one way to solve the inverse problem. Suppose we have a diffusion process $\{\mathbf{x}(t)\}_{t=0}^{T}$ generated by perturbing $\mathbf{x}$ with an SDE, and a

time-dependent score-based model $\mathbf{s}_{\boldsymbol{\theta}*}(\mathbf{x}(t), t)$ trained to approximate $\nabla_{\mathbf{x}} \log p_t(\mathbf{x}(t))$. Once we have an estimate of $\nabla_{\mathbf{x}} \log p_t(\mathbf{x}(t) \mid \mathbf{y})$, we can simulate the reverse-time SDE in Eq. (48) to sample from $p_0(\mathbf{x}(0) \mid \mathbf{y}) = p(\mathbf{x} \mid \mathbf{y})$. To obtain this estimate, we first observe that

$$\nabla_{\mathbf{x}} \log p_t(\mathbf{x}(t) \mid \mathbf{y}) = \nabla_{\mathbf{x}} \log \int p_t(\mathbf{x}(t) \mid \mathbf{y}(t), \mathbf{y}) p(\mathbf{y}(t) \mid \mathbf{y}) \mathrm{d}\mathbf{y}(t),$$

where $\mathbf{y}(t)$ is defined via $\mathbf{x}(t)$ and the forward process $p(\mathbf{y}(t) \mid \mathbf{x}(t))$. Now assume two conditions:

- $p(\mathbf{y}(t) \mid \mathbf{y})$ is tractable. We can often derive this distribution from the interaction between the forward process and the SDE, like in the case of image imputation and colorization.
- $p_t(\mathbf{x}(t) \mid \mathbf{y}(t), \mathbf{y}) \approx p_t(\mathbf{x}(t) \mid \mathbf{y}(t))$. For small $t$, $\mathbf{y}(t)$ is almost the same as $\mathbf{y}$ so the approximation holds. For large $t$, $\mathbf{y}$ becomes further away from $\mathbf{x}(t)$ in the Markov chain, and thus have smaller impact on $\mathbf{x}(t)$. Moreover, the approximation error for large $t$ matter less for the final sample, since it is used early in the sampling process.

Given these two assumptions, we have

$$\begin{aligned}
\nabla_{\mathbf{x}} \log p_t(\mathbf{x}(t) \mid \mathbf{y}) &\approx \nabla_{\mathbf{x}} \log \int p_t(\mathbf{x}(t) \mid \mathbf{y}(t)) p(\mathbf{y}(t) \mid \mathbf{y}) \mathrm{d}\mathbf{y} \\
&\approx \nabla_{\mathbf{x}} \log p_t(\mathbf{x}(t) \mid \hat{\mathbf{y}}(t)) \\
&= \nabla_{\mathbf{x}} \log p_t(\mathbf{x}(t)) + \nabla_{\mathbf{x}} \log p_t(\hat{\mathbf{y}}(t) \mid \mathbf{x}(t)) \\
&\approx \mathbf{s}_{\boldsymbol{\theta}*}(\mathbf{x}(t), t) + \nabla_{\mathbf{x}} \log p_t(\hat{\mathbf{y}}(t) \mid \mathbf{x}(t)),
\end{aligned} \tag{50}$$

where $\hat{\mathbf{y}}(t)$ is a sample from $p(\mathbf{y}(t) \mid \mathbf{y})$. Now we can plug Eq. (50) into Eq. (48) and solve the resulting reverse-time SDE to generate samples from $p(\mathbf{x} \mid \mathbf{y})$.

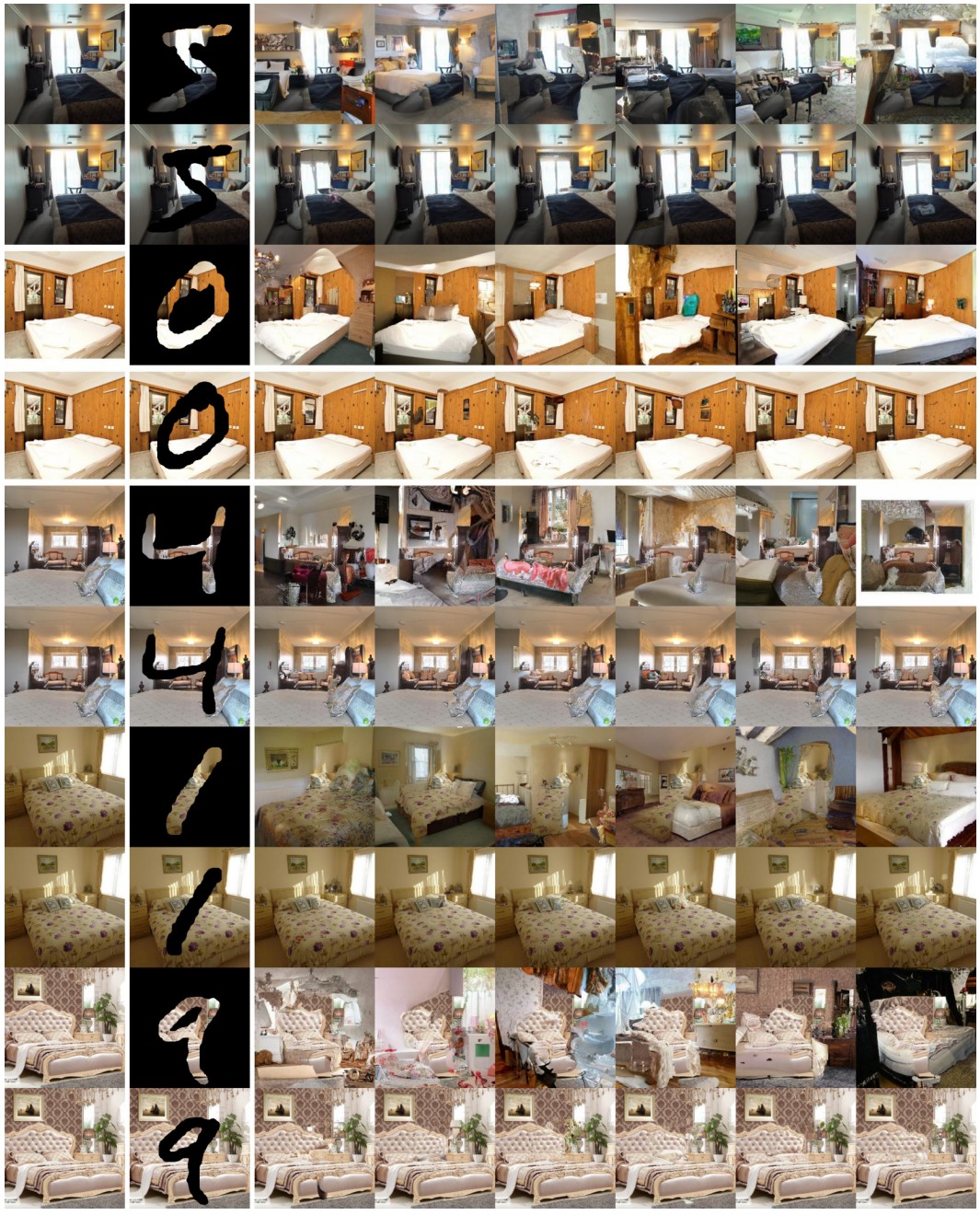

Figure 14: Extended inpainting results for $256 \times 256$ bedroom images.

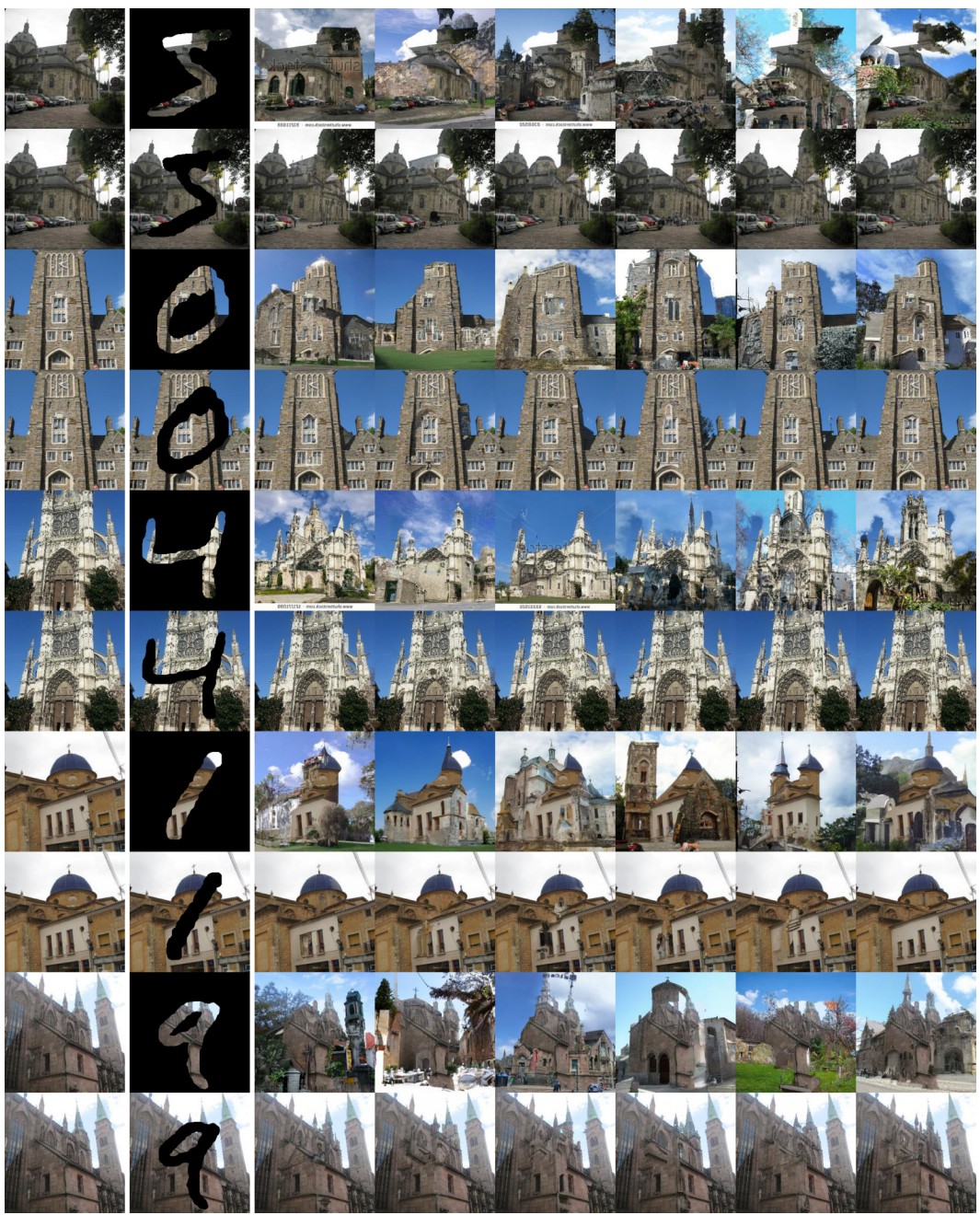

Figure 15: Extended inpainting results for $256 \times 256$ church images.

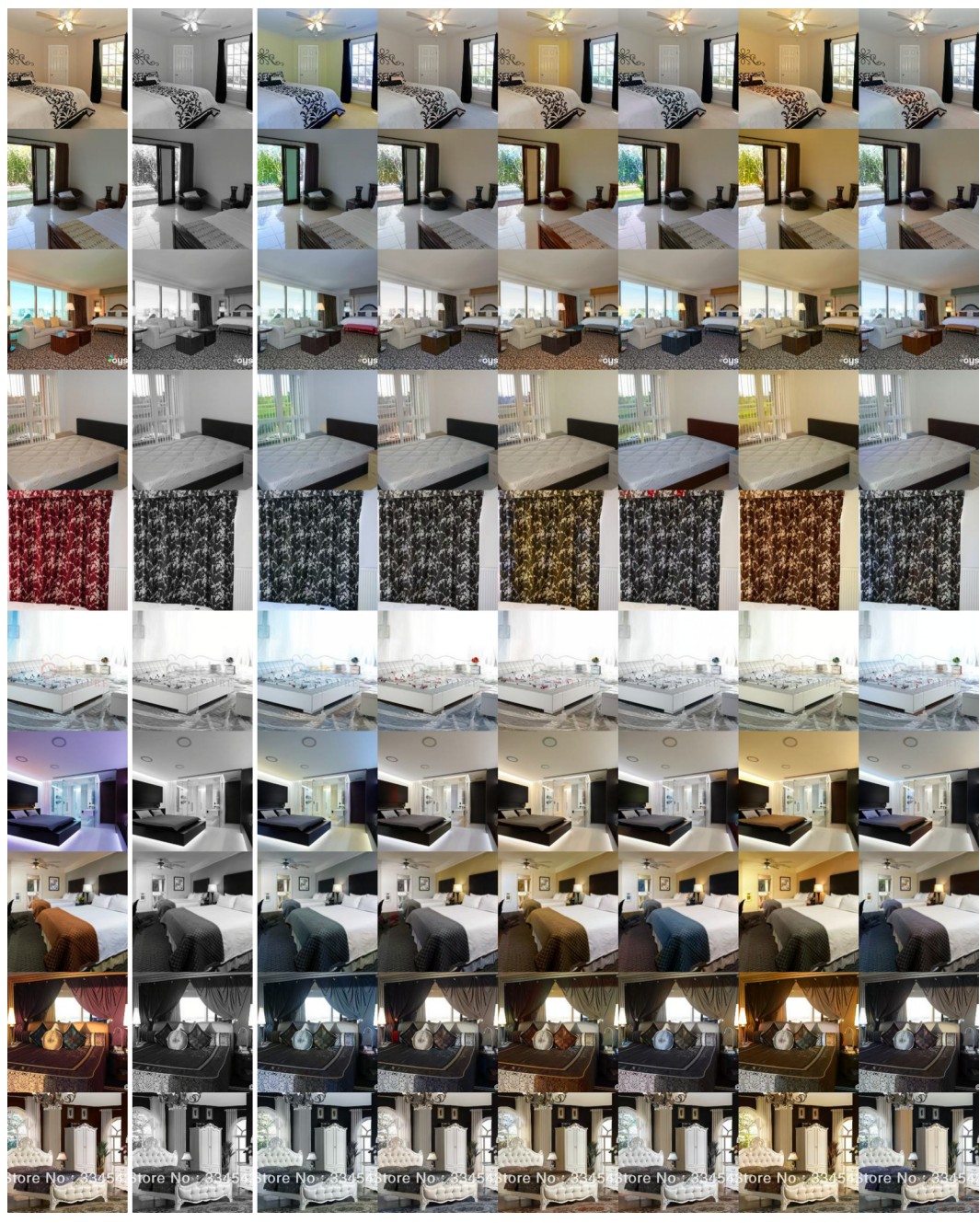

Figure 16: Extended colorization results for $256 \times 256$ bedroom images.

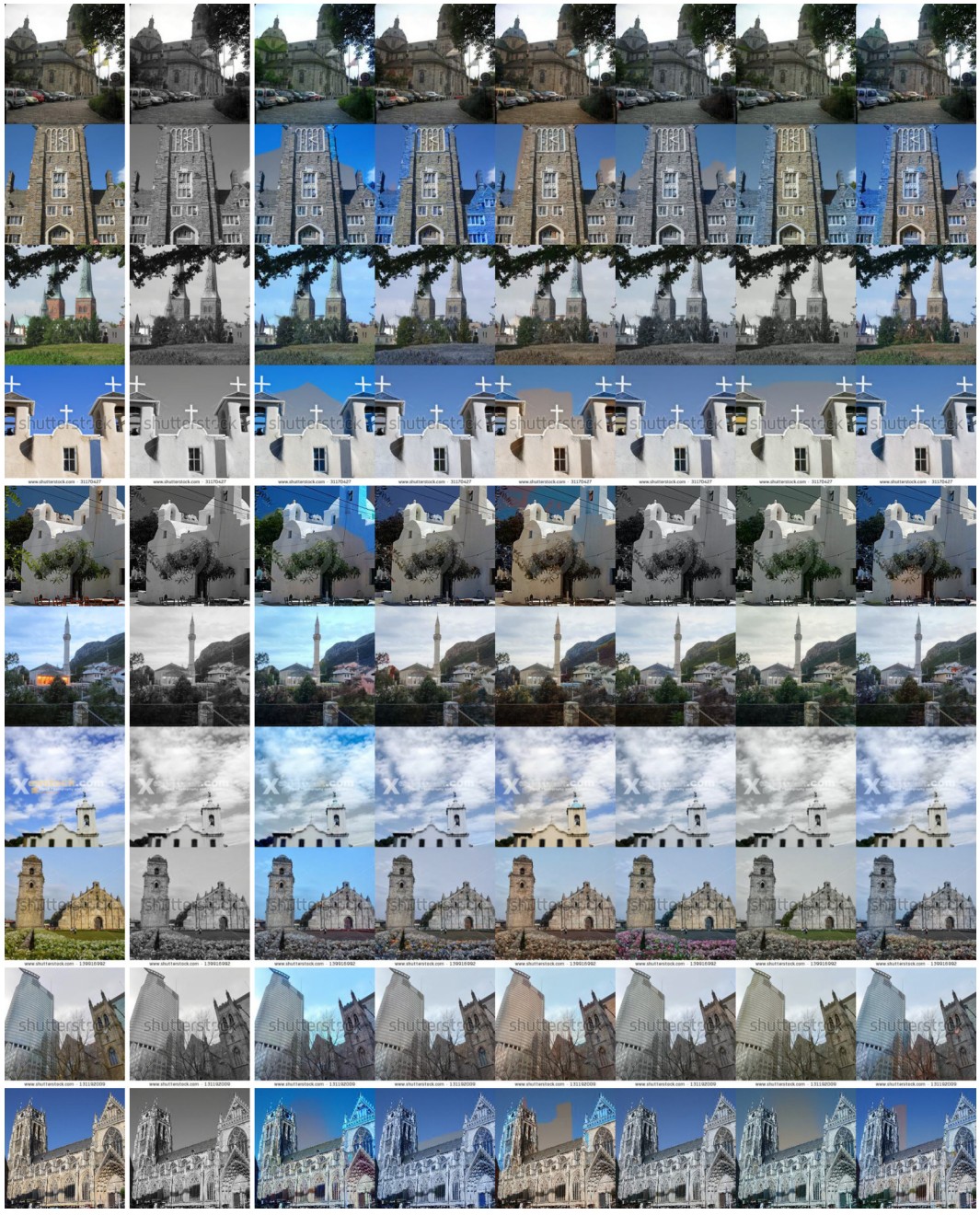

Figure 17: Extended colorization results for $256 \times 256$ church images.

