# OpenReview forum: "Score-Based Generative Modeling through Stochastic Differential Equations"
_ICLR.cc/2021/Conference — ICLR 2021 Oral_

### Official Review · AnonReviewer3 · 2020-10-24
**Official Blind Review #3**

**Rating:** 8
**Confidence:** 4

**Review:**

#### Summary and contributions
This paper proposes a generalized framework for score-based generative modeling (SBGM). The proposed method subsumes previous SBGM techniques of score matching with Langevin dynamics (SMLD aka NCSN) and denoising diffusion probabilistic modeling (DDPM) and shows how they correspond to different discretizations of Stochastic Differential Equations (SDEs). The  continuous-time SDE generalizes the idea of a finite number of perturbation kernels used by previous methods to a continuum of them. The authors propose a forward SDE that transforms the data distribution into a known noise distribution and the corresponding reverse-time SDE that converts samples from this noise distribution to the data distribution. A predictor-corrector sampling framework is studied that leads to improved performance of both NCSN and DDPM frameworks. The paper also shows the equivalence of the proposed SDE to Neural ODEs which allows exact computation of the log-likelihood using the continuous change of variables formula. Quantitative experiments on the CIFAR10 dataset show that the proposed framework leads to significant improvements over previous SBGMs. Qualitative results on the CelebA-HQ dataset demonstrate the ability of the method to scale to high resolution images.

#### Strengths
This paper makes significant technical and empirical contributions to the emerging area of score-based generative models. The generalized SDE framework subsumes recent works in this area and is also connected to Neural ODEs, enjoying exact likelihood calculation, which may be relevant to the normalizing flows and generative modeling community is general. The empirical evaluation is particularly well-done. It bridges the gap between the performance of NCSN and DDPM models leading to state-of-the-art performance. The authors also demonstrate the ability of the method to generate high quality images of human faces when trained on CelebA-HQ dataset. Preliminary experiments on class conditional generation, imputation, and image colorization demonstrate the wide applicability of the proposed method.

#### Weaknesses
The paper does not suffer from any obvious weaknesses. The quantitive experiments could be strengthened by the addition of results on another dataset but the empirical evaluation is sufficient in its current state.

#### Additional feedback
Questions:
- In equation 11, how is the weighting function $\lambda$ chosen?
- In equation 11, apart from being able to sample from the transition kernel, it should also have a closed-form density for the evaluation of the score. Is my understanding correct?
- In equation 21, should there be a discretization step-size corresponding to $\Delta t$?
- In table 1 (a), why does SMLD with corrector only perform so poorly? As far as I understand it is equivalent to NCSN. Can the authors clarify if I misunderstood something?

-------

Post Rebuttal: I thank the authors for clarifying on my questions and updating the manuscript.

---

> ### Author Response · Authors · 2020-11-24
> **Thank you for the questions and feedback**
>
> Thank you for the detailed review and thoughtful feedback. Below we address specific questions.
>
> **Q: In equation 11 (now Eq. (7)), how is the weighting function $\lambda(t)$ chosen?**
>
> A: As in [NCSN](https://arxiv.org/abs/1907.05600) and [DDPM](https://arxiv.org/abs/2006.11239), we recommend using $\lambda \propto 1/\mathbb{E}[|| \nabla_{\mathbf{x}(t)} \log p_{0t}(\mathbf{x}(t) \mid \mathbf{x}(0)) ||_2^2]$. We have updated Section 3.3 to include this.
>
> **Q: In equation 11 (now Eq. (7)), apart from being able to sample from the transition kernel, it should also have a closed-form density for the evaluation of the score. Is my understanding correct?**
>
> A: You are right when using denoising score matching in Eq. (11) (now Eq. (7)). However, If we replace denoising score matching with other score matching objectives like sliced score matching, we only need to sample from the transition kernel and do not have to know its closed form. We have updated the text in Section 3.3 to make this more clear.
>
> **Q: In equation 21 (now Eq. (40)), should there be a discretization step-size corresponding to $\Delta t$?**
>
> A: We absorbed $\Delta t$ into the notations of $\mathbf{f}_i$ and $\mathbf{G}_i$. We have updated Appendix E to make this more clear.
>
> **Q: In table 1 (a), why does SMLD with corrector-only perform so poorly? As far as I understand it is equivalent to NCSN. Can the authors clarify if I misunderstood something?**
>
> A: Yes, SMLD with the corrector-only sampler is equivalent to NCSN. We find that running one Langevin step per noise level yields much worse performance than running many steps per noise level. For example, by using two Langevin steps per noise scale we can improve the FID from 39.2 to 20.4. Increasing the number of steps per noise level improves results and may match NCSN performance but would greatly increase the overall computation cost.

---

### Official Review · AnonReviewer4 · 2020-10-26
**Interesting generative model and good paper**

**Rating:** 7
**Confidence:** 3

**Review:**

SUMMARY

The submission proposes a score-based generative model, which uses an SDE to map the
data distribution to a simple noise distribution and the corresponding reverse-time SDE to
generate samples by mapping the noise to the data space. The proposed model builds
upon and generalises two existing models (SMLD and DDPM) by transforming the data
using a continuous SDE dynamics as opposed to perturbing the data with a finite number
of noise distributions utilized by these models.

##################################################################

REASON FOR SCORE

The paper provides a clear motivation for the proposed modifications to SMLD and DDPM,
as well as a clear technical description of these modifications, their analysis and discussion.
I think the proposed model and sampling algorithms offer substantial conceptual
improvements to the existing models and should be of interest to the community. The
paper is well written and structured.

##################################################################

PROS

+ Clear motivation for the work.
+ Detailed technical description of the proposed models.
+ Interesting discussion of similarities and differences between SMLD, DDPM, and the SDE
based model, as well as corresponding sampling algorithms.
+ Extensive experiments.

##################################################################

CONS

- I found the discussion of equivalent neural ODE and its differences to reverse SDE
 a bit short, especially given that it is used in multiple experiments.
- The case of using general SDEs (not only those derived from SMLD and DDPM) is
mentioned only briefly, leaving it unclear if using a general SDE would require
relatively simple changes, or if the proposed model is limited to SDEs derived from
SMLD and DDPM.

##################################################################

QUESTIONS and COMMENTS

- Is it correct that the function \sigma(t) in Eq. (6) is assumed to be monotonic
and bounded by \sigma_max, while \beta(t) in Eq. (8) is bounded by 1, but doesn't have to
be monotonic?

- In the case of general SDE for noise perturbations in Eq. (9), are there any assumptions on
drift and diffusion function (such as monotonicity or boundedness)?

- In the case of SDEs (6) and (8), \nabla_x p_{0t} (x(t)) is available in closed-form. Is it always
necessary to have such a closed form expression in order to compute the objective (11), or
can it be estimated somehow without it? (I guess for a general SDE, there is typically no
closed-form \nabla_x p_{0t} (x(t)) available)

- Why do you think the FID values for the PC sampler with corrector are higher than without it
for VP SDE? (Table 1b)

- In section 4.2: "[...] PC samplers significantly outperform the corrector-only method, and can
improve over predictor-only approaches for most cases without extra computation." Why does
full PC sampler (with predictor and corrector) not incur extra computation in comparison to
prediction-only approaches. Don't we need to evaluate the approximate score function
s_\theta(x, i) for each of the M steps in the corrector sampler?

- In section 4.3: "[...] deterministic process whose trajectories induce the same evolution
of densities". Does it mean that ODE (12) and reverse SDE (10) map the same noise
distribution p(x(T)) to the same distribution in the data space? If so, are there any
advantages of using a reverse SDE instead of equivalent ODE if the latter is easier to solve
numerically and admits an exact computation of likelihoods?

---

> ### Author Response · Authors · 2020-11-24
> **Thank you for the questions and comments**
>
> Thank you for the detailed review and thoughtful feedback. Below we address specific questions.
>
> **Q: The discussion of equivalent neural ODE and its differences to reverse SDE is a bit short.**
>
> A: Thanks for pointing this out. We have added a new section in the appendix (Appendix D) to expand on probability flow ODEs, where we included a detailed description of the derivation (Appendix D.1), likelihood computation (Appendix D.2), and probability flow sampling (Appendix D.3 and D.4). In addition, we provided additional experiments to verify the uniquely identifiable encoding property in Appendix D.5.
>
> **Q: The case of using general SDEs (not only those derived from SMLD and DDPM) is mentioned only briefly.**
>
> A: We have revised the paper to put more weight on the general SDE. In particular, we reordered Section 3 and moved the discussion of general SDEs before the introduction of VE/VP SDEs. Our general framework in Section 3.1-3.3 can be readily applied to SDEs $d\mathbf{x} = \mathbf{f}(\mathbf{x}, t)dt + g(t) d\mathbf{w}$, when $\mathbf{f}(\mathbf{x},t)$ is affine in $\mathbf{x}$. With a slight modification, we can also use **SDEs of almost any form**, which we discuss in Appendix A. Although we didn’t experiment on other SDEs, the fact that VP and VE SDEs behave differently indicates that exploring other SDEs is likely a valuable future direction.
>
> **Q: Is it correct that the function $\sigma(t)$ in Eq. (6) is assumed to be monotonic and bounded by $\sigma_\text{max}$, while $\beta(t)$ in Eq. (8) is bounded by 1, but doesn't have to be monotonic?**
>
> A: Yes, $\sigma(t)$ in Eq. (6) (now Eq. (9)) is monotonic and bounded by $\sigma_\text{max}$. For $\beta(t)$ in Eq. (8) (now Eq. (11)), it doesn’t have to be monotonic. Although $0 < \beta_i < 1$, the function $\beta(t)$ is not bounded by 1. This is because $\beta(\frac{i-1}{N-1}) = (N-1) \beta_i$ (see Appendix B), where $N$ denotes the number of noise scales.
>
> **Q:  Are there any assumptions on drift and diffusion function (such as monotonicity or boundedness)?**
>
> A: Yes. In order for the SDE to have a unique strong solution, we require both the drift and diffusion function to be globally Lipschitz in both state and time. We have added this assumption to Section 3.1 (right below Eq. (5)).
>
> **Q: Is it always necessary to have a closed form expression for  $\nabla_\mathbf{x} \log p_{0t} (\mathbf{x}(t) \mid \mathbf{x}(0))$ in order to compute the objective (11) (now (7)), or can it be estimated somehow without it?**
>
> A: Not really. When using sliced score matching instead of denoising score matching in Eq. (11) (now Eq. (7)), we only need to sample from $p_{0t}(\mathbf{x}(t) \mid \mathbf{x}(0))$, which can be easily accomplished by simulating the SDE trajectories with numerical SDE solvers (see Appendix A).
>
> When restricted to denoising score matching, we need to know more about $p_{0t}(\mathbf{x}(t) \mid \mathbf{x}(0))$. For SDEs with affine drift functions (like the VE and VP SDEs), $p_{0t}(\mathbf{x}(t) \mid \mathbf{x}(0))$ is always a Gaussian. The mean and variance of this Gaussian can be obtained by solving a system of ODEs and are often in closed-form. For more general SDEs, $p_{0t}(\mathbf{x}(t) \mid \mathbf{x}(0))$ can be obtained in theory by solving the Fokker-Planck equation (Kolmogorov’s forward equation). We have added this discussion to Section 3.3. No matter whether we solve for $p_{0t}(\mathbf{x}(t) \mid \mathbf{x}(0))$ in closed-form or with numerical methods, we only need to do it once. The solution can be reused when applying the same SDE to different datasets or training score-based models with different architectures. We consider developing faster and better estimators for $p_{0t}(\mathbf{x}(t) \mid \mathbf{x}(0))$ an important direction for future research.

---

> > ### Author Response · Authors · 2020-11-24
> > **Continued**
> >
> > **Q: Why do you think the FID values for the PC sampler with corrector are higher than without it for VP SDE? (Table 1) Why does a full PC sampler (with predictor and corrector) not incur extra computation in comparison to prediction-only approaches?**
> >
> > A: We have updated Table 1 with new results and better presentation (see C in [this response](https://openreview.net/forum?id=PxTIG12RRHS&noteId=SGRrXQmYuQs)). For both VP and VE SDEs, adding one corrector step always helps (see PC1000 vs. P1000) but also doubles computation.
> >
> > To match the computation, predictor-only methods need to evaluate the score-based model at more noise scales than those seen at training, which can be accomplished by interpolating between noise scales in an ad hoc way. The performance of predictor-corrector vs. predictor-only with doubled noise scales (PC1000 vs. P2000) depends on many factors, such as the predictor, SDE, and interpolation methods. We experimented with two interpolation strategies, the first leads to results in Table 1 and the latter leads to the newly-added Table 4. For VE SDE, predictor-corrector uniformly outperforms predictor-only with doubled noise scales. For VP SDE results in Table 1, even though predictor-corrector has slightly worse performance than predictor-only for some predictors, it can achieve better performance than all other samplers when using the probability flow predictor; while for VP SDE results in Table 4, predictor-corrector samplers are uniformly better.
> >
> > These results indicate that predictor-corrector methods (PC1000) can be beneficial for both SDEs to various extent, even compared to predictor-only methods with doubled noise scales and matched computation (P2000).
> >
> >
> > **Q: Does it mean that ODE (12) and reverse SDE (10) (now Eq. (6)) map the same noise distribution $p_T(\mathbf{x}(T))$ to the same distribution in the data space? What’s the advantage of reverse SDE?**
> >
> > A: Yes. Theoretically they map the same noise distribution to the same data distribution. In practice solving the reverse SDE for sampling may lead to better performance. For example, in Table 2, the black-box ODE sampler obtained an FID of 3.77 (DDPM (probability flow)), whereas the ancestral sampling sampler (a predictor-only sampler that solves the reverse VP SDE) obtained an FID of 3.17 (DDPM ($L_\text{simple}$)). In general, we empirically find that more accurately solving the reverse SDE or ODE does not always lead to improvements in sample quality and different approximate sampling schemes (such as including a corrector) could potentially improve performance.

---

### Official Review · AnonReviewer2 · 2020-10-27
**Fantastic Paper!**

**Rating:** 9
**Confidence:** 4

**Review:**

Summary: This paper presents a generative model based on stochastic differential equations (SDEs), which generalizes two other score-based generative models score matching with Langevin dynamics (SMLD) and denoising diffusion probabilistic modeling (DDPM). The recipe to sample from the data distribution is based on (i) the observation that both SMLD and DDPM can be formulated as the discretization of an SDE, (ii) the finding from Anderson (1982) about the reverse of an Ito process, (iii) a score model. A novel aspect of the presented technique is the use of the score model as "predictor", which gives the initial sample from the MCMC sampler that serves as "corrector". Finally, the Ito process induced by the reverse SDE is formulated in a deterministic manner, leading to a neural-ODE based generative model.

Pros:
- The authors did a good job at showing connections between the previous score-based generative models and their model. I believe on its own this is a nice contribution.
- The method is thoroughly analyzed. I went through the derivations and didn't find any errors.
- Experiments show that combining the predictor and corrector routines leads to better performance, a nice validation of the theoretical claims.
- As promised, the model achieves SOTA on several tasks.

Cons:
- I'm having difficulty seeing the transformation of the reverse SDE into an ODE (from eq.10 to eq.12). Is it as simple as multiplying the second term with 1/2 and discarding the Brownian motion? Also, eq.12 is a simple ODE system, which has nothing to with a process as far as I understand. Maybe more explanation or pointers in Maoutsa et al., 2020 would be nice.
- The paper lacks the discussion on the benefits/downsides of different SDE solvers, discretization time steps, etc.
- As such, the paper lacks a "toy example" experiment, for example, on a simple 2D dataset like half-moons. A visual demonstration of the SDEs and probability flow (maybe corresponding vector fields and Brownian motion over time) would be interesting.

Additional comments:
- A typo (intead) right below eq.9
- Best performing rows can be bold in Table 1.

---

> ### Author Response · Authors · 2020-11-24
> **Thank you for the feedback**
>
> Thank you for the detailed review and thoughtful feedback. Below we address specific questions.
>
> **Q: How to transform the reverse SDE into the probability flow ODE?**
>
> A: We have included a detailed derivation in Appendix D.1. The proof is based on the Fokker-Planck equation. The probability flow ODE defines a deterministic process that has the same marginal probability densities as the SDE, and thus can be used to generate samples from the same distribution as the SDE.
>
> **Q:  The discussion on the benefits/downsides of different SDE solvers, discretization time steps.**
>
> A: In this paper we mainly focus on demonstrating the flexibility of sampling algorithms allowed by our SDE framework. We compare different sampling methods (which correspond to different discretizations) in Table 1 and Table 2, and demonstrate qualitatively in Figure 4 that probability flow with black-box ODE solvers can save 90% computation without significantly hurting sample quality. We agree with the reviewer that a more detailed quantitative analysis on the merits and limitations of various samplers are important research questions, and consider it to be a major direction for future work.
>
> **Q: The paper lacks a "toy example" experiment.**
>
> A: Great suggestion! We added a toy example in Figure 2 to illustrate the main components of our framework, and demonstrated the trajectories of SDEs and probability flow ODEs.
>
> **Q: Typos and writing suggestions.**
>
> A: Thanks for pointing these out! We have fixed the typo and bolded best performing rows in Table 1.

---

> > ### Comment · AnonReviewer2 · 2020-11-24
> > **Thanks for the response!**
> >
> > The paper looks even better at the moment. I also liked Figure-2 very much in the sense that both stochastic/deterministic forward/backward flows are visualized. Great contribution!

---

### Official Review · AnonReviewer1 · 2020-10-28

**Rating:** 8
**Confidence:** 3

**Review:**

This paper generalizes a family of score-based generative models that rely on sequences of noise scalings of the data and extends them to the continuous domain, which leads to an SDE-based framework. By using score-matching, the forward SDE, which transforms data into a tractable noise distribution, can be reversed and thus used as a generative model. This is then improved by employing a two-phase algorithm with a prediction step, followed by a tunable number of correction steps. Further, reformulating the problem as a neural ODE allows for exact likelihood computations and reduces the number of required function evaluations. The framework enables unconditional, as well as conditional samples.

I find the paper to be very well written and straightforward. As someone who does not have neural SDEs or Langevin Samplers as a core competence, I was able to follow all of the writeup, which is remarkable. I think the framework is nice and there is substantial novel innovation to justify accepting this paper. The experiments are convincing.

A few questions and remarks:
- You claim that you unify current methods into a common framework. While I see that you attempt to do this (i.e. putting the algorithms side-by-side, etc.), but in essence, you still handle VE and VP SDEs separately throughout. My suggestion would be to either really try to unify them into a single formulation, or alternatively, tone down the claims of unification, maybe just say that you show commonalities.
- In Figure 2, you claim that the results are best when computation is "split" between the predictor and corrector. However, this is a very imprecise statement. An equal split would mean just 1 step of corrector, but I don't see clear evidence that that's best. Do you have numerical evidence that a 1-to-1 split is best, or what do you mean by "split"? Otherwise, you could just say that M is a tunable hyperparameter.
- Also in Figure 2, it seems that there is a clear shift at some point where the samples go from low to high quality. Do you have any numerical indication (without looking at a test set, FID, etc.) of how a practicioner could notice that running for more steps would or wouldn't help?
- In Table 1, what is the last row? I guess it's just employing the corrector, but maybe a label for the row would be nice.
- Also in Table 1, it looks like the corrector helps for VE SDEs, but hurts for VP SDEs. Do you have an explanation for this? Maybe it's somewhere in the text, but if it is, I've missed it, so maybe point me to it.
- Given that you can compute exact likelihoods, is it possible to compute the exact NLL for any real dataset, like a test dataset?

---

> ### Author Response · Authors · 2020-11-24
> **Thank you for the questions and feedback**
>
> Thank you for the detailed review and thoughtful feedback. Below we address specific questions.
>
> **Q: To what degree the framework unifies previous approaches. VE and VP SDEs are handled separately throughout.**
>
> A: SMLD and DDPM, the two previous approaches, are both discretizations of SDEs (VE and VP SDE respectively) in our framework. Their original sampling algorithms are special cases of our predictor-corrector sampler. Specifically, SMLD uses an empty predictor and a Langevin dynamics corrector. DDPM uses an ancestral sampling predictor and an empty corrector.
>
> We would like to clarify that VE and VP SDEs are just two instantiations of our SDE-based framework of generative modeling. Although VE and VP SDEs result in seemingly different sampling algorithms, they are powered by the same idea at the core. The predictor-corrector samplers for VE and VP SDEs in Algorithm 1 and 2 both consist of a reverse diffusion predictor and a Langevin dynamics corrector, where the former can be derived by following the same general procedure in Appendix E, and the latter is shared.
>
> To improve clarity, we have reordered Section 3 and put the general framework before the discussion of VE/VP SDEs. The section for VE/VP SDEs is now titled “Special cases: VE and VP SDEs” to emphasize their relationship to the general framework.
>
> **Q: Results are best when computation is split between the predictor and corrector?**
>
> A: By “split”, we mean that computation is allocated to both the predictor and corrector and not to one alone. We agree an even split may not be optimal, and $M$ should be tuned for the best performance. We have rephrased our text according to your suggestion.
>
> **Q: Numerical indication of how a practitioner could notice that running for more steps would or would not help?**
>
> A: In the current work we tune the number of steps by visually inspecting the generated samples. We agree that a numerical indication would be very useful for tuning the sampling algorithms and is a very interesting direction for future research.
>
> **Q: What is the last row in Table 1? I guess it's just employing the corrector.**
>
> A: You are correct, it is the corrector-only sampler. We have re-organized the table to make this more clear.
>
> **Q: In Table 1, it looks like the corrector helps for VE SDEs, but hurts for VP SDEs.**
>
> A: We have updated Table 1 with new results and better presentation (see C in [this response](https://openreview.net/forum?id=PxTIG12RRHS&noteId=SGRrXQmYuQs)). For both VP and VE SDEs, adding one corrector step always helps (see PC1000 vs. P1000) but also doubles computation.
>
> To match the computation, predictor-only methods need to evaluate the score-based model at more noise scales than those seen at training, which can be accomplished by interpolating between noise scales in an ad hoc way. The performance of predictor-corrector vs. predictor-only with doubled noise scales (PC1000 vs. P2000) depends on many factors, such as the predictor, SDE, and interpolation methods. We experimented with two interpolation strategies, the first leads to results in Table 1 and the latter leads to the newly-added Table 4. For VE SDE, predictor-corrector uniformly outperforms predictor-only with doubled noise scales. For VP SDE results in Table 1, even though predictor-corrector has slightly worse performance than predictor-only for some predictors, it can achieve better performance than all other samplers when using the probability flow predictor; while for VP SDE results in Table 4, predictor-corrector samplers are uniformly better.
>
> These results indicate that predictor-corrector methods (PC1000) can be beneficial for both SDEs to various extent, even compared to predictor-only methods with doubled noise scales and matched computation (P2000).
>
> **Q: Is it possible to compute the exact NLL for any real dataset, like a test dataset?**
>
> A: Yes. Our results in Table 2 are NLL numbers on the CIFAR-10 test dataset. We can compute the exact NLL for any input data, just like normalizing flow models and neural ODEs. We have made this point more clear in Section 4.3.

---

### Author Response · Authors · 2020-11-24
**A summary of updates**

We would like to thank all reviewers for providing high quality reviews and constructive feedback that have improved the paper. We are encouraged that reviewers think our paper “makes significant technical and empirical contributions to the emerging area of score-based generative models” (R3); our proposed model and sampling algorithms “offer substantial conceptual improvements to the existing models” (R4); our empirical evaluation is “extensive” (R4),  “particularly well-done” (R3), “a nice evaluation of theoretical claims” (R2); and our writeup “well-written” (R1, R4), “well-structured” (R4) and “was able to follow all without having neural SDEs or Langevin Samplers as a core competence” (R1).

We have updated our draft to further improve the writing and incorporate suggestions from reviewers, extended the appendix with more details for reproducibility, and will be releasing code and model checkpoints. Below, we summarize changes made in the updated submission.

### A. New toy example figure

We have added a new figure (Fig. 2 to Section 3) as suggested by R2. It depicts the SDE and ODE trajectories for transforming a one-dimensional toy data distribution to a standard Gaussian. We believe it provides a good overview of our score-based generative modeling framework with SDEs, and contrasts the trajectories of the SDE and ODE.

### B. Clarifying how our general SDE framework encapsulates VE/VP SDE.

In response to R1 and R4 on how our framework unifies VE and VP SDEs, we have improved the text in Section 3. We now present our general SDE framework before the introduction of VE and VP SDEs, and make it more clear that they are two particular instantiations.

### C. Improved comparison of different samplers

We have improved the clarity of Table 1, provided additional results, and highlighted the computational cost of each sampler. These results show that adding 1 corrector step for each predictor step (PC1000) doubles computation but always improves performance (against P1000). Moreover, it is typically better than doubling the number of predictor steps (P2000), where we have to interpolate between noise scales in an ad hoc manner (detailed in Appendix G) for SMLD/DDPM models. Depending on the interpolation method, predictor-corrector methods may uniformly outperform predictor-only samplers with the same computation (newly-added results in Table 4, PC1000 vs. P2000).

We also improved the results of corrector-only samplers (C2000) by using 2 corrector steps per noise scale, instead of 1 corrector step with interpolated noise scales. Predictor-corrector samplers (PC1000) still perform uniformly better than corrector-only ones.

### D. Additional details on probability flow ODEs

We have polished the writing of Section 4.3. In addition, we added Appendix D, where we provide a detailed derivation of the probability flow ODE (Appendix D.1), full description of likelihood computation (Appendix D.2), detailed description of sampling with probability flows (Appendix D.3 and D.4), as well as additional experimental results for uniquely identifiable encoding (Appendix D.5, Figs. 8 and 9).

### E. Updating results of previous work in Table 2 and 3.

StyleGAN2-ADA authors updated their FID and inception scores after our paper submission, and we have incorporated these new results into Table 3. Our results are still state-of-the-art, though the gap is smaller. We also added results from NCSNv2 to Table 2, and results of RealNVP and iResNet to Table 3.

### F. Inpainting and colorization on $256\times 256$ images.

We have included inpainting and colorization results on $256\times 256$ images from LSUN bedroom and church_outdoor. We updated Section 5, Fig. 5 and provided extended samples in Figs. 13-16 in Appendix I. These results present the same qualitative findings as the original submission with CIFAR-10 figures, but highlight that the method works well on higher resolution images too!

---

### Comment · ~Yang_Zhao5 · 2021-01-18
**Code**

Nice work! When can we expect the code release?

---

> ### Author Response · Authors · 2021-01-20
> **Code is coming soon!**
>
> Thanks for your interest! We are refactoring code and going through Google's code releasing policy. If everything goes well we will release code and checkpoints in the next few days. We will announce code link in a separate comment in openreview.

---

### Author Response · Authors · 2021-02-09
**Code release**

Our code and checkpoints are released at GitHub:
1. JAX + FLAX codebase (recommended): https://github.com/yang-song/score_sde
2. PyTorch codebase: https://github.com/yang-song/score_sde_pytorch

We additionally provide several colab notebooks to help people use our codebase and understand the basic ideas of this paper:
* JAX + FLAX: Load our pretrained checkpoints and play with sampling, likelihood computation, and controllable synthesis. [link](https://colab.research.google.com/drive/1dRR_0gNRmfLtPavX2APzUggBuXyjWW55?usp=sharing)

* PyTorch: Load our pretrained checkpoints and play with sampling, likelihood computation, and controllable synthesis. [link](https://colab.research.google.com/drive/17lTrPLTt_0EDXa4hkbHmbAFQEkpRDZnh?usp=sharing)

* Tutorial of score-based generative models in JAX + FLAX. [link](https://colab.research.google.com/drive/1SeXMpILhkJPjXUaesvzEhc3Ke6Zl_zxJ?usp=sharing)

* Tutorial of score-based generative models in PyTorch. [link](https://colab.research.google.com/drive/120kYYBOVa1i0TD85RjlEkFjaWDxSFUx3?usp=sharing)

---

### Comment · ~Haohang_Xu1 · 2022-07-14
**Some questions about Alg. 5**

Thanks for your nice work. I have some questions about Algorithm 5 in the paper. I can not get why the step size $\epsilon$ of the corrector in the VP SDE setting need to be multiplied by the $\alpha_i$, which does not appear in the VE SED setting.

---

### Decision · Program_Chairs · 2021-01-07
**Final Decision**

**Decision:**

Accept (Oral)

**Comment:**

All reviewers agree that this is a well-written and interesting paper that will be of interest to the ICLR and broader ML community.